# Positively selected modifications in the pore of TbAQP2 allow pentamidine to enter *Trypanosoma brucei*

Ali H Alghamdi[1], Jane C Munday[1], Gustavo Daniel Campagnaro[1], Dominik Gurvic[2], Fredrik Svensson[3], Chinyere E Okpara[4], Arvind Kumar[5], Juan Quintana[6], Maria Esther Martin Abril[1], Patrik Milić[1], Laura Watson[1], Daniel Paape[1], Luca Settimo[1], Anna Dimitriou[1], Joanna Wielinska[1], Graeme Smart[1], Laura F Anderson[1], Christopher M Woodley[4], Siu Pui Ying Kelly[1], Hasan MS Ibrahim[1], Fabian Hulpia[7], Mohammed I Al-Salabi[1], Anthonius A Eze[1], Teresa Sprenger[8], Ibrahim A Teka[1], Simon Gudin[1], Simone Weyand[8], Mark Field[6,9], Christophe Dardonville[10], Richard R Tidwell[11], Mark Carrington[8], Paul O'Neill[4], David W Boykin[5], Ulrich Zachariae[2], Harry P De Koning[1]*

[1]Institute of Infection, Immunity and Inflammation, University of Glasgow, Glasgow, United Kingdom; [2]Computational Biology Centre for Translational and Interdisciplinary Research, University of Dundee, Dundee, United Kingdom; [3]IOTA Pharmaceuticals Ltd, St Johns Innovation Centre, Cambridge, United Kingdom; [4]Department of Chemistry, University of Liverpool, Liverpool, United Kingdom; [5]Chemistry Department, Georgia State University, Atlanta, United States; [6]School of Life Sciences, University of Dundee, Dundee, United Kingdom; [7]Laboratory for Medicinal Chemistry, University of Ghent, Ghent, Belgium; [8]Department of Biochemistry, University of Cambridge, Cambridge, United Kingdom; [9]Institute of Parasitology, Biology Centre, Czech Academy of Sciences, Ceske Budejovice, Czech Republic; [10]Instituto de Química Médica - CSIC, Madrid, Spain; [11]Department of Pathology and Lab Medicine, University of North Carolina at Chapel Hill, Chapel Hill, United States

*For correspondence:
Harry.De-Koning@glasgow.ac.uk

**Abstract** Mutations in the *Trypanosoma brucei* aquaporin AQP2 are associated with resistance to pentamidine and melarsoprol. We show that TbAQP2 but not TbAQP3 was positively selected for increased pore size from a common ancestor aquaporin. We demonstrate that TbAQP2's unique architecture permits pentamidine permeation through its central pore and show how specific mutations in highly conserved motifs affect drug permeation. Introduction of key TbAQP2 amino acids into TbAQP3 renders the latter permeable to pentamidine. Molecular dynamics demonstrates that permeation by dicationic pentamidine is energetically favourable in TbAQP2, driven by the membrane potential, although aquaporins are normally strictly impermeable for ionic species. We also identify the structural determinants that make pentamidine a permeant although most other diamidine drugs are excluded. Our results have wide-ranging implications for optimising antitrypanosomal drugs and averting cross-resistance. Moreover, these new insights in aquaporin permeation may allow the pharmacological exploitation of other members of this ubiquitous gene family.

**eLife digest** African sleeping sickness is a potentially deadly illness caused by the parasite *Trypanosoma brucei*. The disease is treatable, but many of the current treatments are old and are becoming increasingly ineffective. For instance, resistance is growing against pentamidine, a drug used in the early stages in the disease, as well as against melarsoprol, which is deployed when the infection has progressed to the brain. Usually, cases resistant to pentamidine are also resistant to melarsoprol, but it is still unclear why, as the drugs are chemically unrelated.

Studies have shown that changes in a water channel called aquaglyceroporin 2 (TbAQP2) contribute to drug resistance in African sleeping sickness; this suggests that it plays a role in allowing drugs to kill the parasite. This molecular 'drain pipe' extends through the surface of *T. brucei*, and should allow only water and a molecule called glycerol in and out of the cell. In particular, the channel should be too narrow to allow pentamidine or melarsoprol to pass through. One possibility is that, in *T. brucei*, the TbAQP2 channel is abnormally wide compared to other members of its family. Alternatively, pentamidine and melarsoprol may only bind to TbAQP2, and then 'hitch a ride' when the protein is taken into the parasite as part of the natural cycle of surface protein replacement.

Alghamdi et al. aimed to tease out these hypotheses. Computer models of the structure of the protein were paired with engineered changes in the key areas of the channel to show that, in *T. brucei*, TbAQP2 provides a much broader gateway into the cell than observed for similar proteins. In addition, genetic analysis showed that this version of TbAQP2 has been actively selected for during the evolution process of *T. brucei*. This suggests that the parasite somehow benefits from this wider aquaglyceroporin variant.

This is a new resistance mechanism, and it is possible that aquaglyceroporins are also larger than expected in other infectious microbes. The work by Alghamdi et al. therefore provides insight into how other germs may become resistant to drugs.

## Introduction

The *Trypanosoma brucei*-group species are protozoan parasites that cause severe and fatal infections in humans (sleeping sickness) and animals (nagana, surra, dourine) (*Giordani et al., 2016*; *Büscher et al., 2017*). The treatment is dependent on the sub-species of trypanosome, on the host, and on the stage of the disease (*Giordani et al., 2016*; *P. De Koning, 2020*). Many anti-protozoal drugs are inherently cytotoxic but derive their selectivity from preferential uptake by the pathogen rather than the host cell (*Munday et al., 2015a*; *P. De Koning, 2020*). Conversely, loss of the specific drug transporters is a main cause for drug resistance (*Barrett et al., 2011*; *Baker et al., 2013*; *Munday et al., 2015a*; *P. De Koning, 2020*). This is the case for almost all clinically used trypanocides, including diamidines such as pentamidine and diminazene (*Carter et al., 1995*; *de Koning, 2001a*; *de Koning et al., 2004*; *Bridges et al., 2007*), melaminophenyl arsenicals such as melarsoprol and cymelarsan for cerebral stage human and animal trypanosomiasis, respectively (*Carter and Fairlamb, 1993*; *Bridges et al., 2007*), and the fluorinated amino acid analogue eflornithine for human cerebral trypanosomiasis (*Vincent et al., 2010*). The study of transporters is thus important for anti-protozoal drug discovery programmes as well as for the study of drug resistance (*Lüscher et al., 2007*; *Munday et al., 2015a*).

In *Trypanosoma brucei*, the phenomenon of melarsoprol-pentamidine cross-resistance (MPXR) was first described shortly after their introduction (*Rollo and Williamson, 1951*), and was linked to reduced uptake rather than shared intracellular target(s) (*Frommel and Balber, 1987*). The first transporter to be implicated in MPXR was the aminopurine transporter TbAT1/P2 (*Carter and Fairlamb, 1993*; *Mäser et al., 1999*; *Munday et al., 2015b*) but two additional transport entities, named High Affinity Pentamidine Transporter (HAPT1) and Low Affinity Pentamidine Transporter (LAPT1), have been described (*de Koning, 2001a*; *de Koning and Jarvis, 2001*; *Bridges et al., 2007*). HAPT1 was identified as Aquaglyceroporin 2 (TbAQP2) via an RNAi library screen, and found to be the main determinant of MPXR (*Baker et al., 2012*, *Baker et al., 2013*; *Munday et al., 2014*). The apparent permissibility for high molecular weight substrates by TbAQP2 was attributed to the highly unusual selectivity filter of TbAQP2, which lacks the canonical aromatic/arginine (ar/R) and full

NPA/NPA motifs, resulting in a much wider pore (*Baker et al., 2012*; *Munday et al., 2014*; *Munday et al., 2015a*). Importantly, the introduction of TbAQP2 into *Leishmania* promastigotes greatly sensitised these cells to pentamidine and melarsen oxide (*Munday et al., 2014*). Moreover, in several MPXR laboratory strains of *T. brucei* the *AQP2* gene was either deleted or chimeric after cross-over with the adjacent *TbAQP3* gene, which, unlike AQP2, contains the full, classical ar/R and NPA/NPA selectivity filter motifs and is unable to transport either pentamidine or melaminophenyl arsenicals (*Munday et al., 2014*). Similar chimeric genes and deletions were subsequently isolated from sleeping sickness patients unresponsive to melarsoprol treatment (*Graf et al., 2013*; *Pyana Pati et al., 2014*) and failed to confer pentamidine sensitivity when expressed in a *tbaqp2-tbaqp3* null *T. brucei* cell line whereas wild-type TbAQP2 did (*Munday et al., 2014*; *Graf et al., 2015*).

The model of drug uptake through a uniquely permissive aquaglyceroporin (*Munday et al., 2015a*) was challenged by a study arguing that instead of traversing the TbAQP2 pore, pentamidine merely binds to an aspartate residue (Asp265) near the extracellular end of the pore, above the selectivity filter, followed by endocytosis (*Song et al., 2016*). This alternative, 'porin-receptor' hypothesis deserves careful consideration given that (i) it is an exceptional assertion that drug-like molecules with molecular weights grossly exceeding those of the natural substrates, can be taken up by an aquaglyceroporin and (ii) the fact that bloodstream form trypanosomes do have, in fact, a remarkably high rate of endocytosis (*Field and Carrington, 2009*; *Zoltner et al., 2016*). The question is also important because aquaporins are found in almost all cell types and the mechanism by which they convey therapeutic agents and/or toxins into cells is of high pharmacological and toxicological interest. While TbAQP2 is the first aquaporin described with the potential to transport drug-like molecules, this ability might not be unique, and the mechanism by which the transport occurs should be carefully investigated.

We therefore conducted a mutational analysis was undertaken, swapping TbAQP2 and TbAQP3 selectivity filter residues and altering pore width at its cytoplasmic end. This was complemented with a thorough structure-activity relationship study of the interactions between pentamidine and TbAQP2, using numerous chemical analogues for which inhibition constants were determined and interaction energy calculated. The pentamidine-TbAQP2 interactions were further modelled by running a molecular dynamics simulation on a protein-ligand complex, and in addition, we investigated a potential correlation between the *T. brucei* endocytosis rate and the rate of pentamidine uptake. Our results provide strong evidence for pentamidine permeating directly through the central pore of TbAQP2. Having identified the essential characteristics that allow the transport of large, flexible molecules through TbAQP2, this should now allow the evaluation of aquaporins in other species for similar adaptations.

## Results

### Investigation of the structural determinants of AQP2 for pentamidine transport

#### Positive selection for pore size

In *T. brucei*, the AQP2 and AQP3 genes are arranged as a tandem pair on chromosome 10 and have 74% amino acid identity. Whereas TbAQP2 clearly mediates pentamidine uptake, TbAQP3 does not (*Baker et al., 2012*; *Munday et al., 2014*), nor do various chimeric AQP2/3 rearrangements that give rise to pentamidine resistance (*Munday et al., 2014*; *Graf et al., 2015*). To investigate the origin of the AQP2 gene, a phylogenetic analysis of AQPs in African trypanosomes was performed. The number of aquaporin genes varies: there is a single aquaporin in *T. vivax* and *T. congolense,* two in *T. suis* and three in *T. brucei* and its derivatives and the most probable tree (*Figure 1*) is consistent with the evolutionary history of the four species (*Hutchinson and Gibson, 2015*), and indicates AQP1 as the ancestral AQP present in all trypanosome species. A duplication occurred in the common ancestor of *T. suis* and *T. brucei* after divergence from *T. congolense* and a further duplication, to form AQP2 and AQP3, in the ancestor of *T. brucei* after divergence from *T. suis*. Multiple alignment (*Figure 1—figure supplement 1*) shows that the classical NPA/NPA and ar/R AQP selectivity filter elements are present in all AQPs except *T. brucei* AQP2. The divergence of *T. brucei* AQP2 and 3 was investigated by calculating the non-synonymous/synonymous codon ratio (dN/dS) for

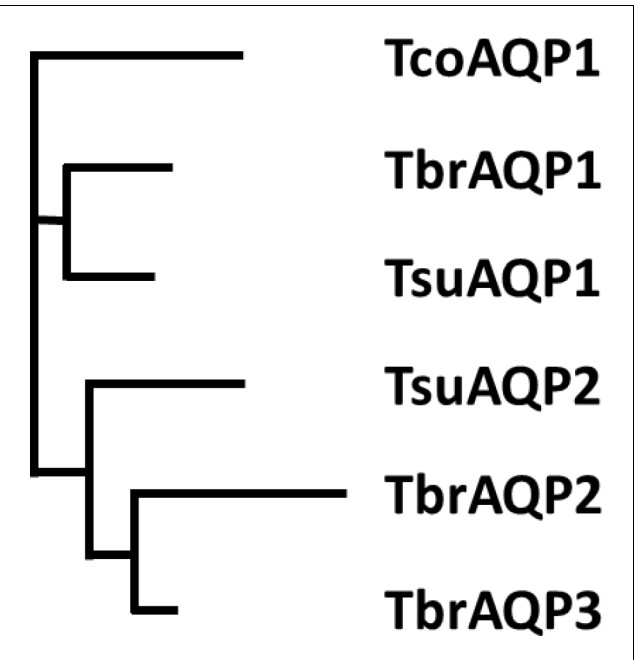

**Figure 1.** Phylogenetic tree of *Trypanosoma* aquaporins. The tree is a Neighbour-joining tree produced in Clustal Omega with the lengths of the horizontals proportional to the differences.

The online version of this article includes the following figure supplement(s) for figure 1:

**Figure supplement 1.** Sequence alignment and individual sequences of the *T. congolense, T. b. brucei* and *T. suis.*

**Figure supplement 2.** The ratio of non-synonymous v synonymous (dN/dS) codon changes calculated for selected comparisons between *T.*

different AQPs (*Figure 1—figure supplement 2*). For *T. brucei* aligned with *T. suis* AQP1, dN/dS is 0.21 and for AQP3, dN/dS is 0.30 indicating purifying selection. However, comparing *T. brucei* AQP2 with *T. brucei* AQP3, dN/dS is 2.0 indicating strong selection pressure for divergence on AQP2 towards an aquaporin with increased pore size. In order to verify any role of amino acids along the TbAQP2 pore in facilitating pentamidine sensitivity and/or uptake, we performed a mutational analysis.

*Figure 1* shows that replacing the AQP2 NSA/NPS motif in the selectivity filter of the pore with the consensus NPA/NPA motif of other aquaporins still allows pentamidine to permeate, although at substantially reduced rate, with an $EC_{50}$ at >50 fold lower than the *aqp2/aqp3 null* control, while cymelarsan is almost completely blocked ($EC_{50}$ ~60% of control). Similar observations are made for the L258Y mutant (*Figure 1B,C*). In contrast, the single mutant L264R did not allow either cymelarsan or pentamidine permeation (*Figure 1A, E*) and the double mutant I110W/L264R actually enabled a level of cymelarsan (but not pentamidine) permeation (*Figure 1F*). These findings can be rationalised by the differing structural features, interaction patterns, and charge of the two compounds. The leucine to arginine substitution introduces a further positive charge into the central pore of the aquaporin, which repels the dicationic pentamidine molecule and is therefore likely to decrease its affinity to this variant, whereas the neutral arsenical compound is much less affected by the addition of positive charge into the pore.

## Introduction of AQP3 residues into the AQP2 selectivity filter

One highly conserved motif of aquaporins, believed to be essential for permeant selectivity, is NPA/NPA which is present in TbAQP3 but not in TbAQP2, where, uniquely, it is $NS^{131}A/NPS^{263}$ instead. We therefore constructed a TbAQP2 variant with the classical NPA/NPA motif (TbAQP2$^{S131P/S263A}$) and expressed it in the *aqp2/aqp3* null cell line (*Baker et al., 2012*; *Munday et al., 2014*). In this cell line, uptake of 30 nM [$^3$H]-pentamidine was reduced to 4.40 ± 0.71% (n = 4) of the rate in the

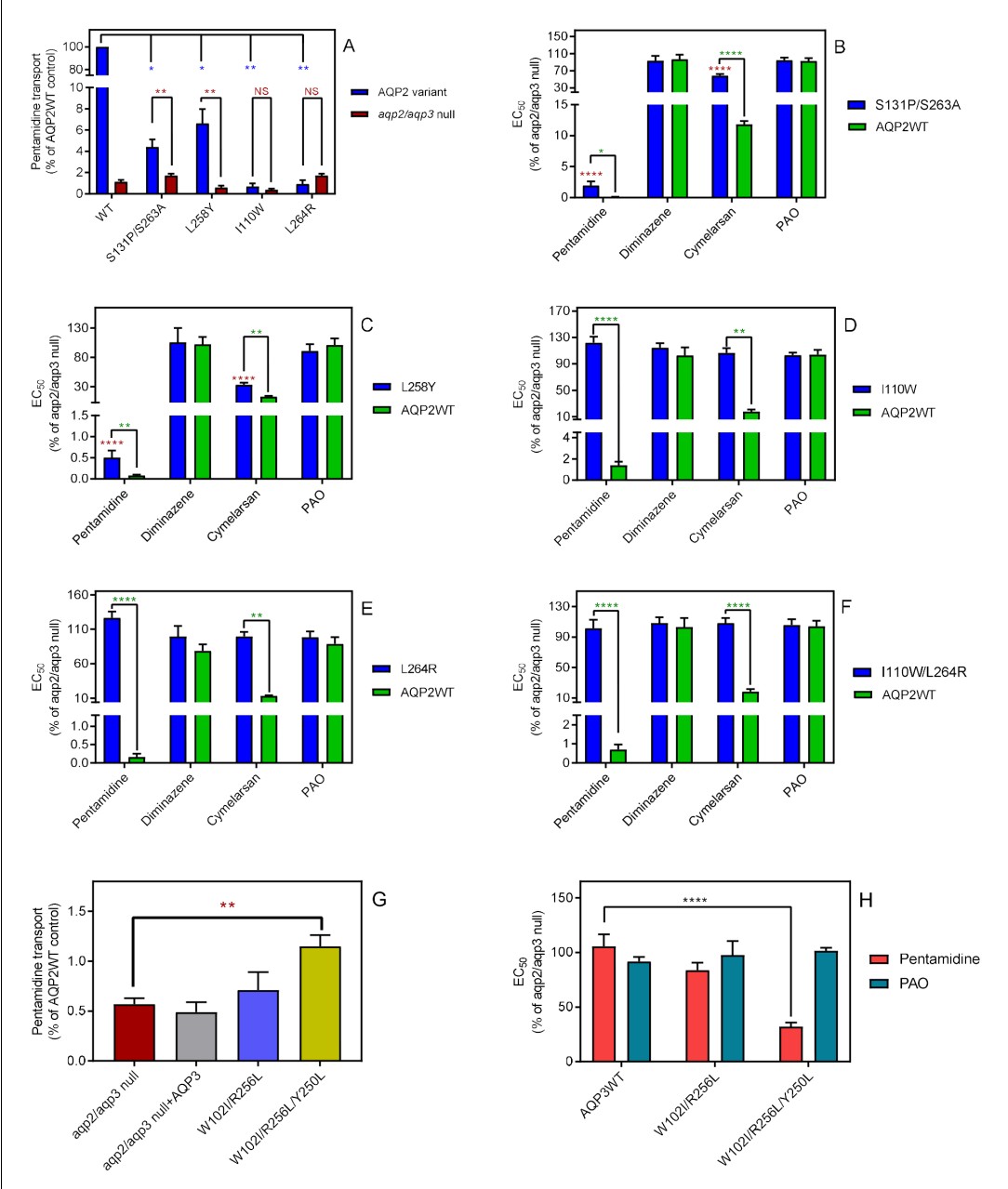

**Figure 2.** The selectivity filter differences between TbAQP2 and TbAQP3 are largely responsible for their differences in pentamidine sensitivity and transport rates. (A) Transport of 30 nM [³H]-pentamidine by *tbaqp2/aqp3* null cells expressing TbAQP2-WT or one of the TbAQP2 mutants as indicated (blue bars). The corresponding brown bars are pentamidine transport in the control *tbaqp2/aqp3* null cells assessed in parallel in each experiment. Transport was determined in the presence of 1 mM adenosine to block the TbAT1/P2 transporter. Bars represent the average and SEM of at least three independent experiments, each performed in triplicate. Blue stars: statistical significance comparison, by two-tailed unpaired Student's tests, between the cells expressing TbAQP2WT and mutants; red stars: statistical comparison between the AQP2-expressing cells and control cells; NS, not significant. (B–F) EC$_{50}$ values indicated test drugs, expressed as a percentage of the resistant control (*tbaqp2/tbaqp3* null), against cell lines either expressing the indicated TbAQP2 mutant or TbAQP2WT (sensitive control). Red stars and green stars: comparison with *tbaqp2/aqp3* null or TbAQP2WT-expressing cells, respectively, which were always assessed in parallel in each experiment. (G) Transport of 30 nM [³H]-pentamidine by *tbaqp2/aqp3* null cells expressing TbAQP3 or an AQP3 mutant as indicated. (H) EC$_{50}$ values of the indicated drugs against *tbaqp2/aqp3* null cells expressing either TbAQP3 or a mutant thereof, expressed as percentage of *tbaqp2/aqp3* null. All data for these graphs are contained in *Figure 2—source data 1*. All experiments are the average and SEM of at least three independent experiments. *, p<0.05; **, p<0.01; ***, p<0.001, ****, p<0.0001 by unpaired Student's t-test, two-tailed.

The online version of this article includes the following source data for figure 2:

**Source data 1.** Individual and average EC$_{50}$ values and transport rates for *Figure 2A-H*.

control line expressing TbAQP2WT (p<0.05, Student's unpaired t-test), as well as significantly different from the rate measured in parallel in the *tbaqp2/tbaqp3* null cells (p<0.01) (**Figure 2A**). The remaining pentamidine uptake was sufficient to strongly sensitise the TbAQP2$^{S131P/S263A}$ cells to pentamidine in a standard protocol of 48 hr incubation with the drug followed by a further 24 hr in the presence of the resazurin indicator dye (p<0.0001 vs *tbaqp2/tbaqp3* null) but the EC$_{50}$ was still significantly higher than the TbAQP2WT control (p<0.05) (**Figure 2B**). A similar effect was observed for the melaminophenyl arsenical drug cymelarsan, but there was no change in sensitivity to diminazene or the control drug phenylarsine oxide (PAO), which is believed to diffuse directly across the membrane (**Fairlamb et al., 1992**; **Figure 2B**).

The mutant L258Y, which has the AQP3 Tyr-250 half of the highly conserved aromatic/arginine (ar/R) motif, responsible for pore restriction and proton exclusion (**Wu et al., 2009**), introduced into the TbAQP2 pore, yielded a drug transport phenotype similar to TbAQP2$^{S131P/S263A}$. The [$^3$H]-pentamidine transport rate was reduced to 6.6 ± 1.4% of TbAQP2WT (p<0.05) but remained above the rate in the *tbaqp2/tbaqp3* null cells (p<0.01) (**Figure 2A**). Pentamidine and cymelarsan EC$_{50}$ values were also significantly different from both the TbAQP2WT and the *tbaqp2/tbaqp3* null controls (**Figure 2C**).

The ar/R motif is part of the larger selectivity filter, usually WGYR, present in both TbAQP1 and TbAQP3 but uniquely consisting of I$^{110}$VL$^{258}$L$^{264}$ in TbAQP2 (**Baker et al., 2013**), all non-polar, open chained residues. Cell lines expressing mutations AQP2$^{I110W}$ and AQP2$^{L264R}$, either alone or in combination, displayed pentamidine transport rates, and pentamidine and cymelarsan EC$_{50}$ values that were not significantly different from the *tbaqp2/tbaqp3* null controls but highly significantly different from the TbAQP2WT drug-sensitive controls, showing that their capacity for pentamidine and cymelarsan uptake had been reduced to zero (**Figure 2A,D–F**).

We conclude that the unique TbAQP2 replacement of the NPA/NPA motif and all of the WGYR selectivity filter mutations are necessary for the observed pentamidine and melaminophenyl arsenical sensitivity observed in cells expressing wild-type TbAQP2.

## Introduction of TbAQP2 selectivity filter residues into the AQP3 pore enables pentamidine transport

An interesting question was whether the introduction of (some of) the critical TbAQP2 residues in TbAQP3 might give the latter the capacity to take up pentamidine. We therefore constructed TbAQP3$^{W102I/R256L}$ and TbAQP3$^{W102I/R256L/Y250L}$ and tested whether *tbaqp2/tbaqp3* null cells transfected with these mutant aquaporins were able to take up 25 nM [$^3$H]-pentamidine in the presence of 1 mM adenosine (which blocks uptake via TbAT1/P2). Pentamidine uptake in the tested cell lines was very low compared to the same cells expressing TbAQP2WT (**Figure 2G**). However, by measuring [$^3$H]-pentamidine uptake over 30 min it was possible to reliably and reproducibly measure radiolabel accumulation in each cell line. This showed that while uptake in TbAQP3$^{W102I/R256L}$ only trended slightly upwards (p>0.05), the mutant AQP3 with all three AQP2 WGYR residues (W102I, R256L and Y250L) accumulated significantly more [$^3$H]-pentamidine than the *tbaqp2/tbaqp3* null cells (p<0.01) or the null cells expressing TbAQP3WT (p=0.011). This is further corroborated by comparing the pentamidine sensitivity profile of these cell lines: only TbAQP3$^{W102I/R256L/Y250L}$ conveyed significant sensitisation to *tbaqp2/tbaqp3* null cells (p<0.0001; **Figure 2H**). Thus, TbAQP3 is converted into a pentamidine transporter by the insertion of the AQP2 WGYR residues, although this does not convey as high a rate of pentamidine uptake as TbAQP2. Similar experiments did not show any sensitisation to cymelarsan upon expression of TbAQP3 or mutants TbAQP3$^{W102I/R256L}$ and TbAQP3$^{W102I/R256L/Y250L}$, although the latter mutant actually showed a 50% higher EC$_{50}$ concentration (p<0.001); EC$_{50}$ values were 114 ± 5%, 105 ± 8% and 159 ± 8% of the *tbaqp2/tbaqp3* control, respectively (n ≥ 7).

## Mutations of amino acids modelled to potentially bind pentamidine or melarsoprol dramatically reduce pentamidine transport

Our previous attempts at modelling the binding of pentamidine and melarsoprol into the pore of TbAQP2 tentatively identified several residues that could be involved in this process (**Munday et al., 2015a**), from which we selected two residues, Ile190 and Trp192, at the extracellular end of the channel (position shown in Figure 7), to swap with the corresponding residues of TbAQP3, creating

TbAQP2$^{I190T}$ and TbAQP2$^{W192G}$. Both residues were predicted to interact with the substrate(s) via main-chain carbonyl oxygen atoms, but the side chains could nonetheless affect the interactions.

TbAQP2$^{I190T}$ displayed dramatically reduced [$^3$H]-pentamidine uptake, at 2.7 ± 0.7% (p<0.01, n = 4) of the TbAQP2WT control, although significantly higher than the rate of the *tbaqp2/tbaqp3* null negative control (p<0.05) (*Figure 3A*). The reduced rate was the result of a reduced $V_{max}$ of the high affinity [$^3$H]-pentamidine uptake, rather than a change in Km; the LAPT1 $V_{max}$ and $K_m$ were unchanged in cells expressing TbAQP2$^{I190T}$ or TbAQP2WT (*Figure 3—figure supplement 1*). TbAQP2$^{I190T}$ still conferred some increased pentamidine sensitivity in the standard resazurin test (p<0.0001), although highly significantly less sensitizing than TbAQP2WT (p<0.001); an intermediate sensitivity was also observed for cymelarsan (*Figure 3B*). Substitution W192G also produced intermediate sensitivity to both drugs (*Figure 3C*) but the double substitution TbAQP2$^{I190T/W192G}$ displayed no significant pentamidine uptake above *tbaqp2/tbaqp3* null (*Figure 3A*) and did not sensitise to pentamidine or cymelarsan (*Figure 3D*).

## The effect of large amino acids at the cytoplasmic end of the pore

To test whether restrictions at the cytoplasmic end of TbAQP2 would impact on pentamidine transport, we selected three leucine residues and exchanged each with tryptophan, creating L84W,

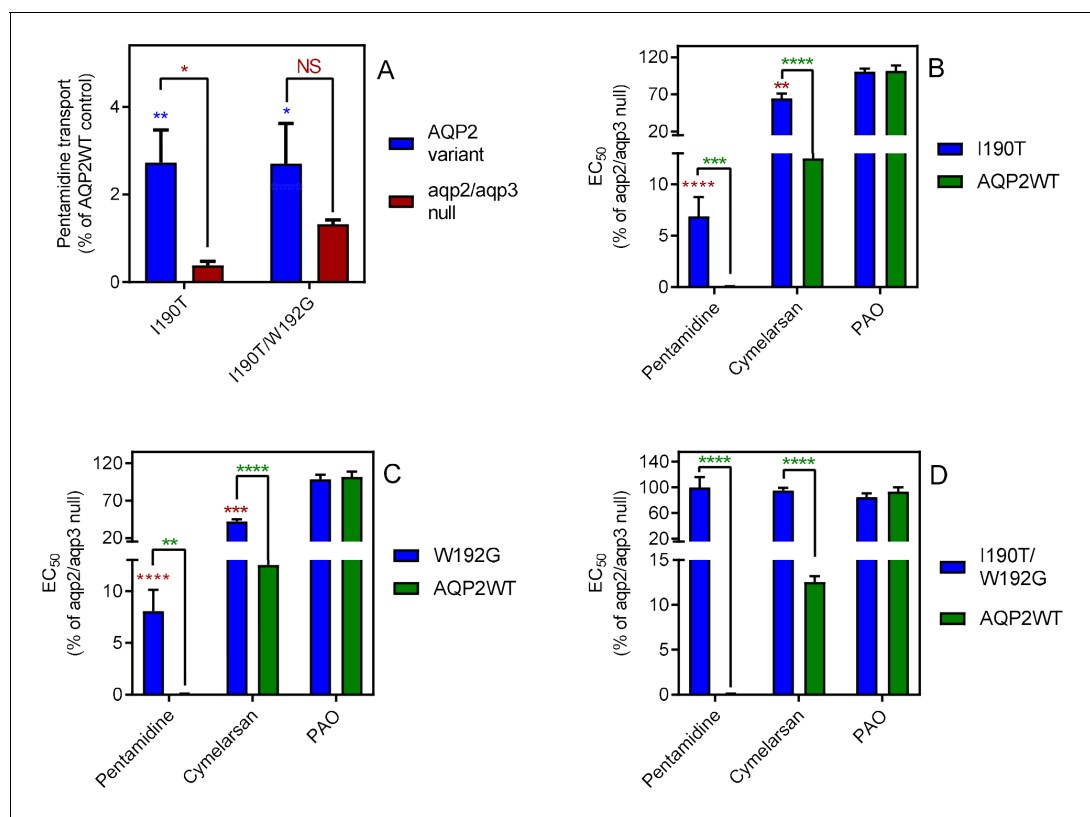

**Figure 3.** Mutational analysis of TbAQP2 residues I190 and W192. (**A**) Transport of 30 nM [$^3$H]-pentamidine by *tbaqp2/tbaqp3* null cells or TbAQP2 variants expressed therein. Transport was expressed as a percentage of the rate of the AQP2WT control, performed in parallel. Blue stars are comparison with TbAQP2WT, red stars, comparison with the *tbaqp2/tbaqp3* null control. NS, not significant. (**B**) EC$_{50}$ values for the indicated drugs against *tbaqp2/tbaqp3* null cells, and against TbAQP2WT and TbAQP2$^{I190T}$ expressed therein; values were expressed as % of the *tbaqp2/tbaqp3* null (resistant) control. Red stars, comparison with the resistant control; green stars, comparison with the internal sensitive control (TbAQP2WT). The assays for all three strains and all three drugs were done simultaneously on at least three different occasions. (**C**) As B but for TbAQP2$^{W192G}$. (**D**) As B but for TbAQP2$^{I190T/W192G}$. All data for these graphs are contained in *Figure 3—source data 1*. *, p<0.05; **, p<0.01; ***, p<0.001, ****, p<0.0001 by unpaired Student's t-test.

The online version of this article includes the following source data and figure supplement(s) for figure 3:

**Source data 1.** Individual and average EC$_{50}$ values and transport rates for *Figure 3A-D* and *Figure 3—figure supplement 1*.
**Figure supplement 1.** Pentamidine transport analysis for TbAQP2$^{I190T}$ and TbAQP2WT.

L118W and L218W (positions indicated in Figure 7). Expressing each of the L-to-W mutants in *tbaqp2/tbaqp3* null cells revealed significantly reduced pentamidine sensitivity compared to the same cells expressing TbAQP2WT (*Figure 4A*), while also exhibiting dramatically reduced rates of [$^3$H]-pentamidine transport (*Figure 4B*). This effect was additive, with TbAQP2$^{L84W/L118W}$ not significantly sensitising for pentamidine and displaying no detectable increase in [$^3$H]-pentamidine transport relative to *tbaqp2/tbaqp3* null cells (*Figure 4A,B*). None of these L-to-W mutants sensitised the cells to cymelarsan, diminazene or PAO, except that the L218W mutant sensitised slightly to diminazene (~2 fold, $p<0.05$; compared to 20-fold for pentamidine, $p<0.0001$) (*Figure 4—figure*

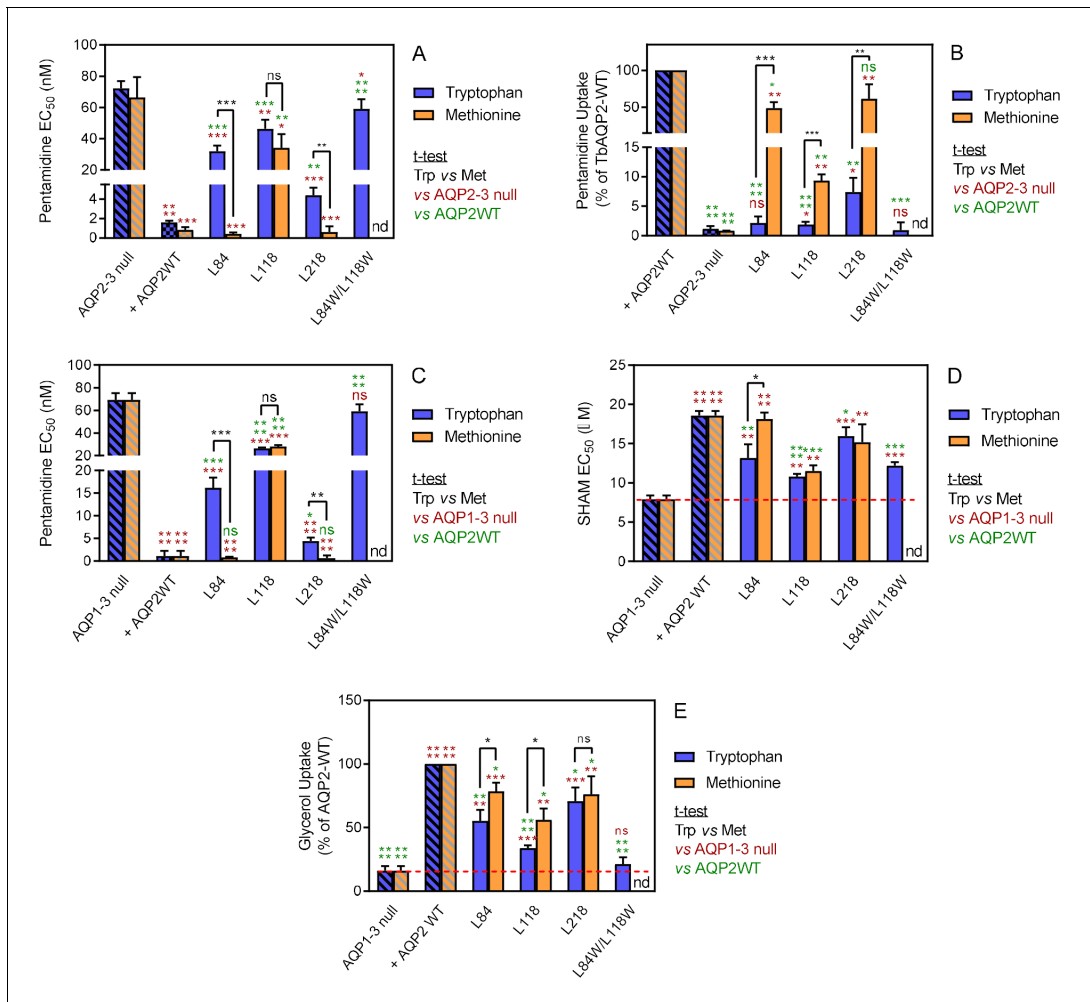

**Figure 4.** Analysis of TbAQP2 variants with a leucine-to-tryptophan or leucine-to-methionine substitution near the cytoplasmic end of the pore. (**A**) Pentamidine EC$_{50}$ values (nM) for mutant and WT TbAQP2 expressed in *tbaqp2/tbaqp3* cells (aqp2-3 null). The mutants are either a Trp (dark blue bars) or Met (orange bars) substitution at the indicated positions. The resistant control (aqp2-3 null) and sensitive control (AQP2WT) for the separate datasets (Trp or Met) are indicated as hatched bars in the same colours. (**B**) As (**A**) but showing transport of 30 nM [$^3$H]-pentamidine by the same cell lines, expressed as percentage of the transport rate in the TbAQP2 control cells. (**C**) Pentamidine EC$_{50}$ values for the same mutants as in (**A**) but expressed in the *tbaqp1-2-3* null cells, performed in parallel with the determination of EC$_{50}$ values for SHAM, shown in (**D**). As all cell lines were done simultaneously, the resistant and sensitive strain control values are identical for the Trp and Met mutants in this series. All bars represent the average and SEM of at least three independent replicates. *, $p<0.05$; **, $p<0.01$; ***, $p<0.001$, ****, $p<0.0001$ by unpaired Student's t-test; ns, not significant; nd, not determined. All data for these graphs are contained in *Figure 4—source data 1*.

The online version of this article includes the following source data and figure supplement(s) for figure 4:

**Source data 1.** Individual and average EC$_{50}$ values and transport rates for *Figure 4A-E* and *Figure 4—figure supplement 1*.
**Figure supplement 1.** EC$_{50}$ values for Cymelarsan, diminazene aceturate and phenylarsine oxide (PAO) against the *tbaqp2-tbaqp3* null cell line.
**Figure supplement 2.** Correlation of the EC$_{50}$ value with the rate of pentamidine transport for all 19 cell lines expressing a wild-type or mutant TbAQP2 in the aqp2/3 null *T. b. brucei* line.

supplement 1). When the same leucine residues were replaced with methionine instead of tryptophan, variants L84M and L218M were not or barely different from TbAQP2WT with respect to pentamidine sensitisation (Figure 4A) or transport (but highly significantly different from their respective tryptophan variants). For position 118 the Met replacement had similar effects as the Trp variant had, albeit with a significantly higher rate of pentamidine transport ($1.88 \pm 0.20$ ($n = 6$) versus $9.38 \pm 0.63\%$ ($n = 3$) of TbAQP2WT, $p<0.001$; Figure 4A,B). The L84M and L218M mutants also sensitised to cymelarsan ($p<0.01$) and, surprisingly, the L218 M mutants also sensitised somewhat to diminazene (~2 fold, $p<0.05$) (Figure 4—figure supplement 1).

These results strongly argue that the introduction of large amino acids at the cytosolic end significantly blocks the transport of pentamidine, whereas the change to Leu→Met mutants were more permissive for pentamidine, but not cymelarsan. In order to check whether these variants were still functional aquaglyceroporins, we used the observation of Jeacock et al., 2017 that T. brucei cells lacking all three AQPs are sensitised to the Trypanosome Alternative Oxidase inhibitor SHAM, as a result of cellular glycerol accumulation. By this measure, all of the position 84, 118 and 218 Trp and Met mutants were able to transport glycerol, as each displayed SHAM EC$_{50}$ values significantly different from the tbaqp1-2-3 null cells (Figure 4C,D); several variants displayed an intermediate SHAM EC$_{50}$, being also significantly different from TbAQP2WT, indicating some attenuation of glycerol efflux capacity for those mutants. Indeed, uptake of [$^3$H]-glycerol closely mirrored the SHAM observations (Figure 4E).

## Overall correlation between [$^3$H]-pentamidine transport rate and pentamidine EC$_{50}$

The results presented in Figures 2–4 consistently show that even TbAQP2 mutants that display a large reduction in [$^3$H]-pentamidine uptake rate results can show intermediate pentamidine sensitivity phenotypes (EC$_{50}$s), due to the nature of the standard drug sensitivity test employed, which involves a 48 hr incubation with the drug prior to a further 24 hr incubation with resazurin: even a much-reduced transport rate will be sufficient to accumulate significant amounts of intracellular pentamidine over 3 days. A plot of [$^3$H]-pentamidine transport rates versus pentamidine EC$_{50}$, using the data for all 19 TbAQP2 and TbAQP3 mutants for which the transport rates were determined, shows that relatively small changes in EC$_{50}$ occur, even with up to approximately 95% reduction in transport rates; at >95% reduction large EC$_{50}$ increases become apparent (Figure 4—figure supplement 2).

## Partially blocking endocytosis does not alter the rate of pentamidine transport and pentamidine does not trigger AQP2 endocytosis in bloodstream form *T. brucei*

The knockdown of the CRK12 kinase in *T. brucei* causes a highly reproducible defect in endocytosis that affects an estimated one third of cells 12 hr after RNAi induction and is ultimately lethal (Monnerat et al., 2013). We utilized this system to investigate whether a link between endocytosis and pentamidine transport exists. At 12 hr of CRK12 RNAi induction with tetracycline, CRK12 mRNA levels were reduced by 42% ($p<0.001$) relative to uninduced controls as determined by qRT-PCR (Figure 5A). Samples from the culture taken at this time point showed an increased abundance of cells with swelling characteristic of endocytosis defects, although this was hard to quantify as a minority of cells were affected, and to various degrees, as the 12 hr time point was deliberately taken at as early a point as possible, so as not to affect cell viability (Figure 5—figure supplement 1) or cause excessive cellular pathology. We thus performed parallel uptake experiments with [$^3$H]-pentamidine and [$^3$H]-suramin, with suramin acting as positive control as it is known to enter *T. brucei* bloodstream forms through endocytosis after binding to surface protein ISG75 (Zoltner et al., 2016; Zoltner et al., 2020). After 12 hr of CRK12 RNAi induction, pentamidine uptake was not significantly less than in the *T. brucei* 2T1 parental cells, whereas uptake of [$^3$H]-suramin was ($p=0.019$, $n = 5$; Figure 5B,C).

As this approach is necessarily limited to a partial inhibition of endocytosis in BSF *T. brucei*, it was also investigated whether pentamidine induces the internalisation and turnover of TbAQP2, as could be expected if the protein acts to internalise substrate by receptor-mediated endocytosis. Cells pretreated with the protein synthesis inhibitor cycloheximide (100 µg/ml) were incubated in the

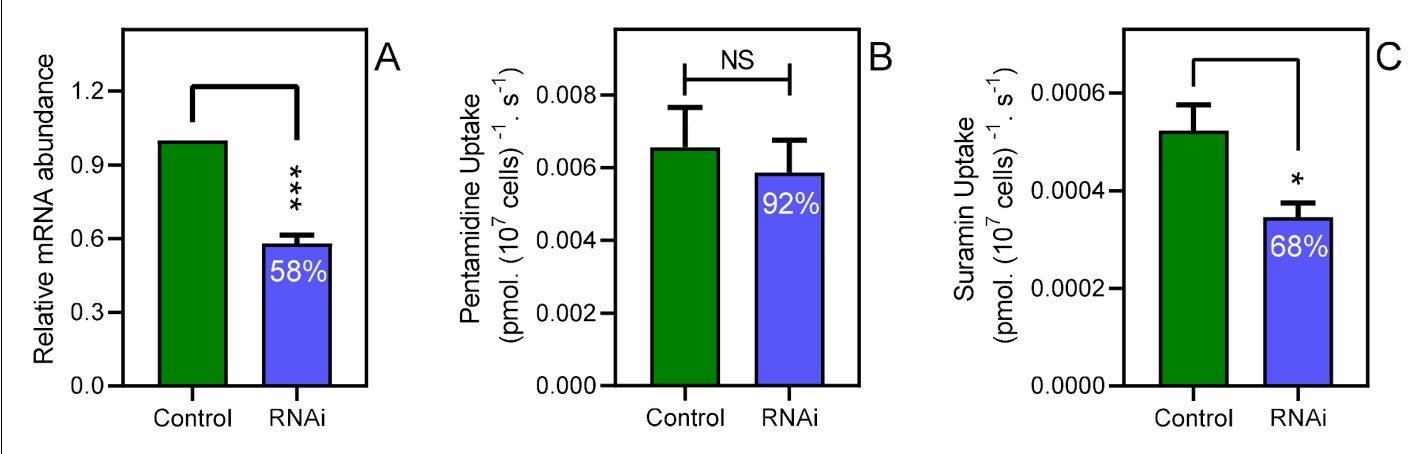

**Figure 5.** Disabling endocytosis does not reduce uptake of pentamidine. (A) qRT-PCR of CRK12, normalised to housekeeping gene GPI-8 (n = 3). (B) Transport of 0.025 µM [³H]-Pentamidine measured in control (non-induced) and CRK12 cell after exactly 12 hr of tetracycline induction; incubation time with label was 30 s. Bar is average and SEM of 5 independent determinations, each performed in triplicate. NS, not significant by unpaired Student's t-test. (C) As frame B but uptake of 0.25 µM [³H]-suramin over 15 min; average and SEM of 5 independent determinations, each in quadruplicate. **, p=0.0027 by Student's unpaired, two-tailed t-test. All data for these graphs are contained in *Figure 5—source data 1*.

The online version of this article includes the following source data and figure supplement(s) for figure 5:

**Source data 1.** mRNA abundance for *Figure 5A* plus transport rates for *Figure 5B, C* and *Figure 5—figure supplement 1*.

**Figure supplement 1.** Growth Curve of CRK12 RNAi cells in full HMI-9 medium at 37°C/5% $CO_2$, in the presence or absence of 1 µg/ml tetracycline (tet).

presence or absence of 25 nM pentamidine and the abundance of 3 × HA tagged TbAQP2 was followed over a period of 6 hr by western blot (*Figure 6—figure supplement 1*). Quantification of the bands showed identical turnover rates with or without pentamidine present in the medium (*Figure 6*).

## The protonmotive force drives AQP2-mediated pentamidine uptake in bloodstream forms of *T. brucei*

It has been reported that knock-down of the HA1–three plasma membrane proton pumps of *T. brucei* (which are essential for maintaining the plasma membrane potential), confers pentamidine resistance (*Alsford et al., 2012*; *Baker et al., 2013*). Interestingly, this locus only conferred resistance to (dicationic) pentamidine, not to the (neutral) melaminophenyl arsenicals, unlike knockdown of the TbAQP2/TbAQP3 locus (*Alsford et al., 2012*). We have previously reported that the HAPT-mediated pentamidine uptake in *T. brucei* procyclics correlates strongly with the proton-motive force (PMF) (*de Koning, 2001a*). However, it is not clear whether this dependency indicates that pentamidine uptake is mediated by a proton symporter, as known for many *T. brucei* nutrient transporters (*de Koning and Jarvis, 1997a*; *de Koning and Jarvis, 1997b*; *de Koning and Jarvis, 1998*; *de Koning et al., 1998*), or reflects the energetics of uptake of cationic pentamidine being driven by the strong inside-negative membrane potential $V_m$. The absence of an effect of HA1–three knockdown on sensitivity to the neutral melaminophenyl arsenicals strongly argues against a mechanism of proton symport for HAPT1/AQP2 but a (partial) dependency of HAPT1/AQP2-mediated uptake of dicationic pentamidine on PMF or $V_m$ would be expected if the substrate traverses the channel, as opposed to binding a single Asp residue on the extracellular side of the protein, as suggested in the endocytosis model (*Song et al., 2016*). Here we show that the same ionophores that inhibit HAPT1-mediated pentamidine transport in procyclic cells, and inhibit hypoxanthine uptake in both bloodstream form (BSF) (*de Koning and Jarvis, 1997b*) and procyclic (*de Koning and Jarvis, 1997a*) *T. brucei*, also dose-dependently inhibit [³H]-pentamidine uptake in BSF (*Figure 7A*). This confirms that pentamidine needs the membrane potential for rapid uptake, as predicted by the dependence on the HA1–three proton pumps. Using [³H]-suramin as an endocytosed substrate (*Zoltner et al., 2016*), we found that 20 µM CCCP also inhibits endocytosis in *T. brucei*, by 32.6% (p=0.029; preincubation 3 min, plus suramin accumulation over 10 min) (*Figure 7B*). While that means that the

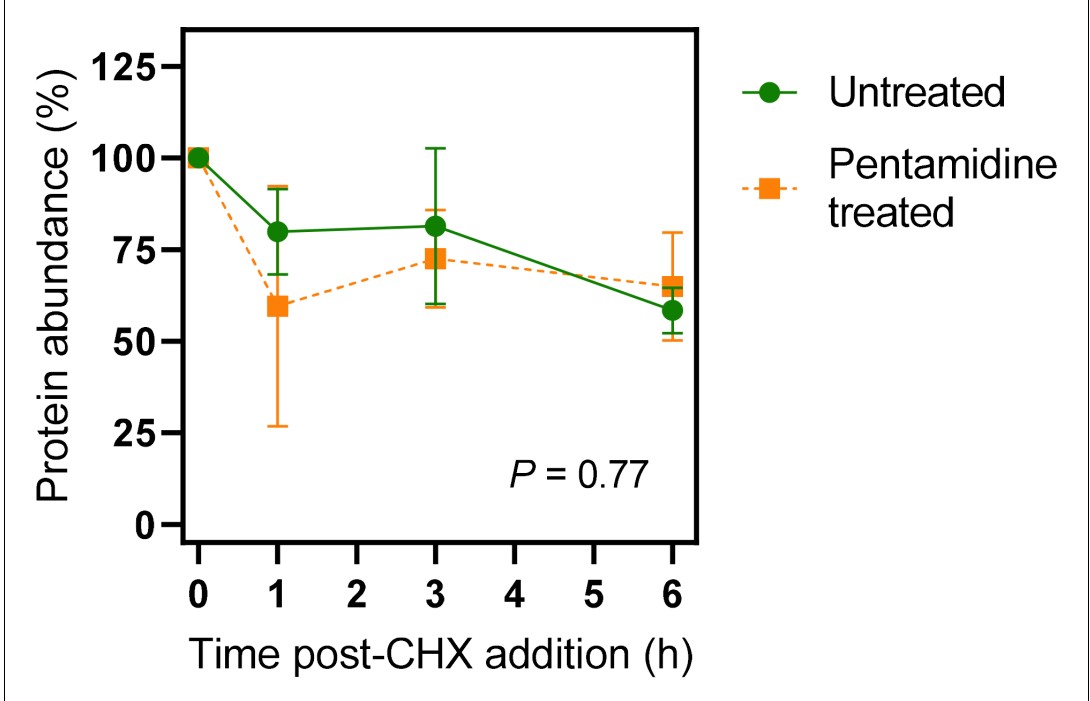

**Figure 6.** Quantification of western blots of [3×HA]TbAQP2. Cells were induced for expression of [3×HA]TbAQP2, pretreated with cyclohexidine and subsequently with 25 nM pentamidine (5× $EC_{50}$). Western blots (*Figure 6—figure supplement 1*) were performed using anti-HA antiserum in order to quantify the relative amount of TbAQP2 in the cells. The two datasets were not significantly different by Kolmogorov-Smirnov test (p=0.77) and data points at each time point were also not significantly different by Student's t-test (p>0.05). All data for these graphs are contained in *Figure 6—source data 1*.

The online version of this article includes the following source data and figure supplement(s) for figure 6:

**Source data 1.** Densitometry readings with Western blotting of TbAQP2 turnover, normalised to 0 h.

**Figure supplement 1.** Western blots for a [3×HA]AQP2 turnover assay in untreated *T. brucei* 2T1 cells, or in the presence of pentamidine 25 nM.

ionophore experiments do not perfectly discriminate between endocytosis and trans-channel transport for di-cationic pentamidine, they do for neutral melaminophenyl arsenicals: the non-dependence of these neutral TbAQP2 substrates on the proton gradient (*Alsford et al., 2012*) indicates that, unlike suramin, they are not endocytosed.

Although there is a good correlation between the proton-motive force and TbAQP2-mediated pentamidine transport (*Figure 7C*), the effect of CCCP was stronger than expected, and stronger than previously observed for [3H]-hypoxanthine uptake in *T. brucei* bloodstream forms (*de Koning and Jarvis, 1997b*) and we thus investigated whether CCCP might have a direct effect on TbAQP2. Indeed, CCCP inhibited uptake of (neutral) [3H]-glycerol in *tbaqp1-2-3* null cells expressing TbAQP2-WT, with an $IC_{50}$ of 20.7 ± 2.6 µM (n = 3) and inhibited [3H]-pentamidine uptake in the same cells with a similar $IC_{50}$ (*Figure 7D,E*), showing CCCP to inhibit TbAQP2 directly, irrespective of effects on the membrane potential. *Figure 7D* also shows that pentamidine, used as a control, inhibits [3H]-glycerol uptake with an $EC_{50}$ value (Mean of 27.5 nM, n = 2) similar to the $EC_{50}$ of pentamidine inhibiting uptake of [3H]-pentamidine.

## Molecular dynamics modelling of pentamidine interactions with TbAQP2

To further investigate pentamidine binding and permeation in TbAQP2, we used the coordinates of the TbAQP2-pentamidine complex that was modelled in our previous study (*Munday et al., 2015a*). The stability of the protein model was first confirmed by unbiased atomistic molecular dynamics simulations (*Figure 8—figure supplement 1*). We then conducted force-probe simulations, in which a moving spring potential was used to enforce unbinding of pentamidine from its docked binding

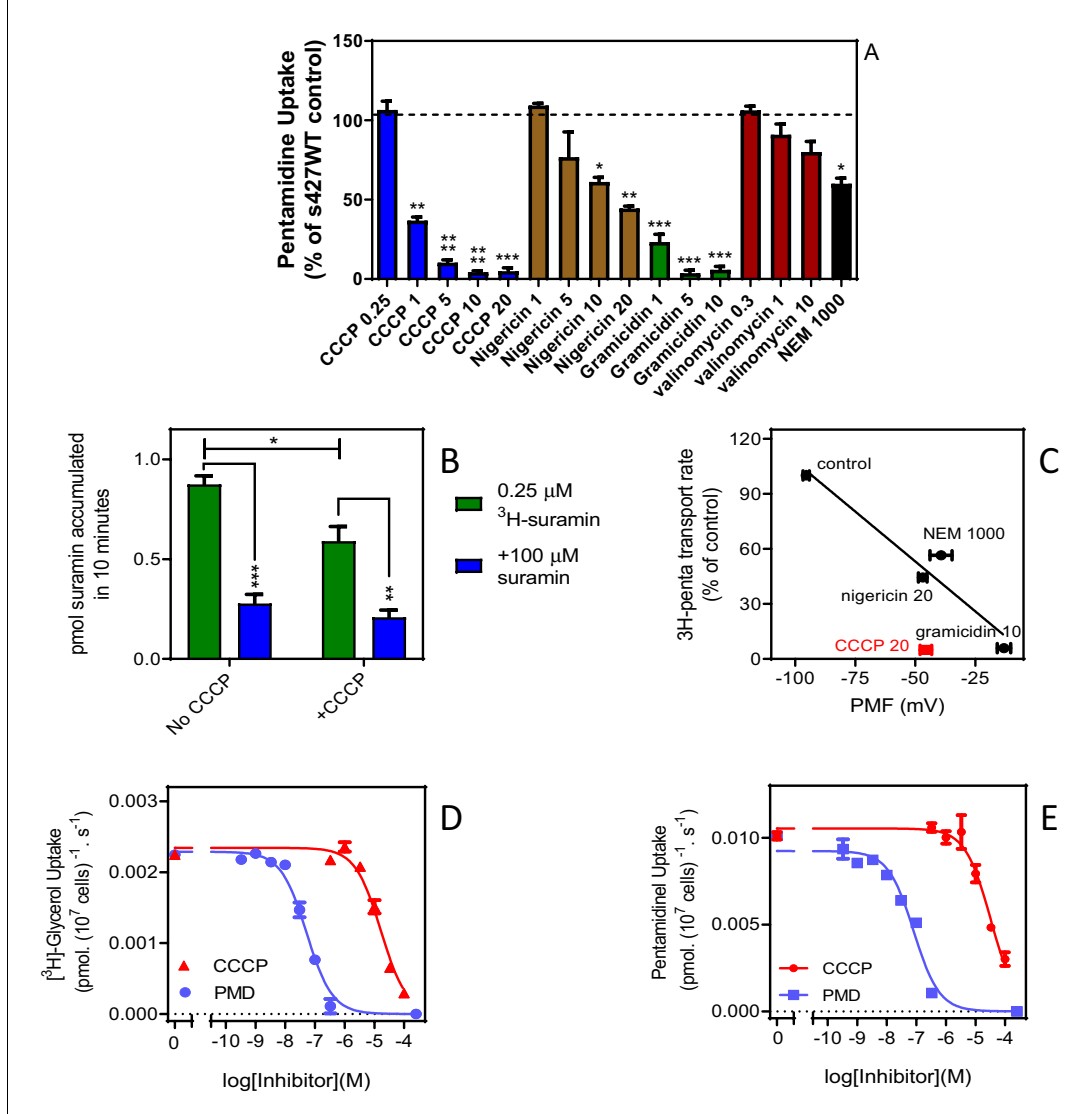

**Figure 7.** High affinity pentamidine uptake in *T. b. brucei* is sensitive to ionophores. (**A**) Uptake of 25 nM [³H]-pentamidine in s427WT bloodstream forms was measured in the presence of 1 mM adenosine to block the P2 transporter, and in the further presence of various ionophores at the indicated concentrations in μM. Incubation with radiolabel was 5 min after a 3 min pre-incubation with ionophore. Accumulation of radiolabel was expressed as a percentage of the control, being a parallel incubation in the absence of any ionophore. Bars represent the average of 3–5 independent determinations (each performed in quadruplicate) and SEM. (**B**) Uptake of 0.25 μM [³H]-suramin by *T. b. brucei* s427WT cells over 10 min. Cells were incubated in parallel, with or without the presence of 20 μM CCCP (plus 3 min pre-incubation). Saturation of the suramin-receptor interaction was demonstrated by including 100 μM unlabelled suramin (blue bars). Bars represent average and SEM or three independent experiments, each performed in quadruplicate. (**C**) Correlation plot of pentamidine transport rate versus protonmotive force (PMF), $r^2$ = 0.93, p<0.05 by F-test. Concentrations in μM are indicated in the frame. CCCP is shown in red and not included in the regression analysis. Each data point is the average of 4 or more independent repeats performed in quadruplicate. The values for PMF were taken from *de Koning and Jarvis, 1997b*. (**D**) Uptake of 0.25 μM [³H]-glycerol by *aqp1/aqp2/ aqp3* null cells expressing TbAQP2-WT. Dose response with CCCP and pentamidine (PMD), using an incubation time of 1 min. The graph shown was performed in triplicate and representative of three independent repeats. (**E**) As D but using 0.025 μM [³H]-pentamidine and 30 s incubations. Representative graph in triplicate from three independent repeats. *, p<0.05; **, p<0.01; ***, p<0.001 by Student's unpaired t-test. All data for these graphs are contained in *Figure 7—source data 1*.

The online version of this article includes the following source data for figure 7:

**Source data 1.** Individual and average transport rates and PMF values for *Figure 7A-E*.

position and subsequently reconstructed the free-energy profile of pentamidine association-dissociation along the pore axis by employing Jarzynski's equality (*Park et al., 2003*).

*Figure 8A* shows that the docked position of pentamidine correctly identified its minimum free-energy binding site inside the TbAQP2 pore. Pentamidine adopts an extended state inside the TbAQP2 pore, adapting its molecular shape to the narrow permeation channel; pentamidine binding poses display inter-amidine lengths in the range 16.5–17 Å. Importantly, our steered simulations reveal that pentamidine can exit the channel in either direction, and that unbinding on the route towards the cytoplasm occurs on a free-energy surface roughly symmetric to that towards the extracellular side. Apart from overcoming the strongly attractive binding interaction in the centre, there are no major further free-energy barriers in either direction. The computed free-energy profile of pentamidine binding to the TbAQP2 structural model slightly overestimates its experimentally recorded binding affinity. However, the pentamidine conformation binding the narrow pore may not be the lowest-energy internal conformation of the small molecule, a factor that may be underrepresented in the profile as simulations were started from the protein-bound state. A further source of uncertainty stems from the protein model, which is expected to be somewhat less accurate than a crystal structure.

Due to the dicationic character of pentamidine, the free-energy profile of the molecule within TbAQP2 strongly depends on the membrane voltage. The voltage drop of −125 mV across the cytoplasmic membrane of *T. b. brucei* (*de Koning and Jarvis, 1997b*), with a negative potential inside the cell, results in an overall inward attraction of ~22 kJ/mol (*Figure 8A*, arrow), that is exit from TbAQP2 into the cytoplasm is substantially more favourable for pentamidine than towards the extracellular side. Taken together, the free-energy profile under membrane voltage explains the strong coupling between pentamidine uptake and $V_m$ observed in the experiments. The high affinity of the binding interaction leads to slow off-rates and a relatively low $V_{max}$ (0.0044 ± 0.0004 pmol(10$^7$ cells)$^{-1}$ s$^{-1}$) (*de Koning, 2001a*).

We further investigated the bound positions of pentamidine and melarsoprol in AQP2 by docking, in order to rationalise the differential behaviour of pentamidine and melaminophenyl arsenicals observed in the studied AQP2 mutants. The binding modes in the central pore of AQP2 obtained by docking are shown in *Figure 8B*. They reveal that both drugs are likely to bind to the same general region within the central pore in spite of their different sizes and charge states.

Conversely, the shorter arsenical agent is more affected by introducing the NPA/NPA motif since its terminal polar function intensely interacts with the NSA/NPS motif, whereas the major interactions of pentamidine are seen outside this motif. In the case of the L258Y mutant, the difference can be attributed to the extended flexible linker region in the centre of pentamidine, which is likely to enable it to bend around the added bulky side chain in the mutant, while cymelarsan lacks this level of flexibility in its centre.

Finally, the single and double mutations I190T and W192G have broadly similar effects on pentamidine and cymelarsan permeation across AQP2 (*Figure 2*) as both positions at the entrance to the pore exhibit similar interaction patterns to both pentamidine and the arsenical agent.

## SAR of the pentamidine-AQP2 interaction

In order to study substrate binding and selectivity by the *T. b. brucei* High Affinity Pentamidine Transporter (HAPT1/TbAQP2), competition assays were performed with a series of pentamidine analogues and other potential inhibitors, in the presence of 1 mM unlabelled adenosine to block diamidine uptake by the TbAT1 aminopurine transporter (*de Koning, 2001a*; *Bridges et al., 2007*). High specific activity [$^3$H]-pentamidine was used at 30 nM, below the $K_m$ value (*de Koning, 2001a*). Uptake was linear for at least 3 min (*de Koning, 2001a*) and we utilized 60 s incubations for the determination of inhibition constants ($K_i$). At 30 nM [$^3$H]-pentamidine there is virtually no uptake through LAPT1 (*Bridges et al., 2007*) ($K_m$ value ~1000 fold higher than HAPT1) (*de Koning, 2001a*). The full dataset of 71 compounds is presented in *Supplementary file 1*, featuring $K_i$s spanning five log units.

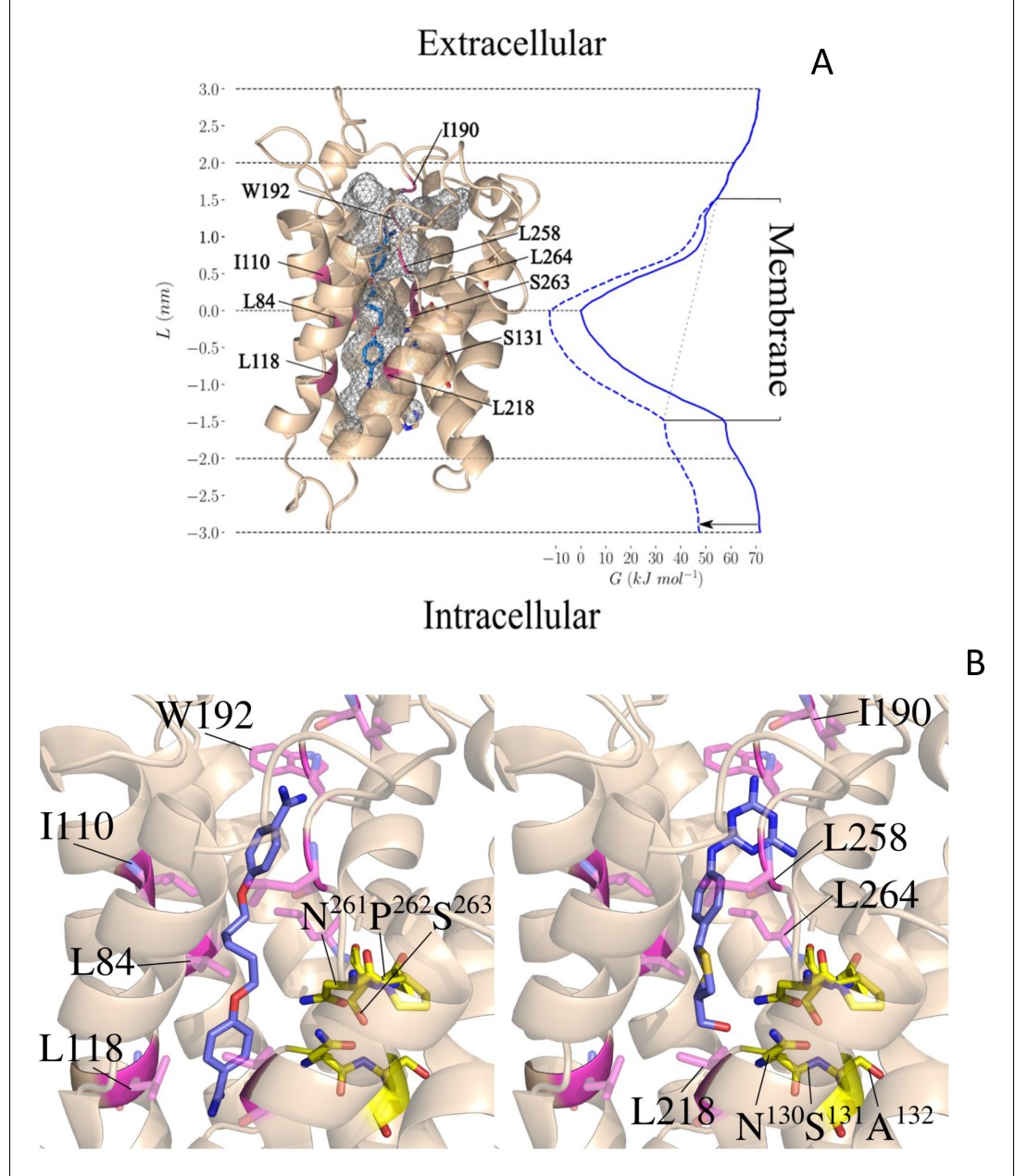

**Figure 8.** Pentamidine binding in TbAQP2 and free-energy profile of permeation (Left). (**A**) Docked conformation of pentamidine (blue) bound to the TbAQP2 (wheat). The protein and the ligand were modelled as described.[4] The protein pore is shown in grey mesh, and the mutated positions described in the text are in magenta. (Right) Free-energy profile G(L) (solid blue line) along the pore axis of TbAQP2 (L). The membrane voltage of *T. b. brucei* gives rise to a voltage drop across the membrane (gray dotted line), which alters the free-energy profile (dashed blue line includes $V_m$ effect) and reduces the free-energy of pentamidine exit into the intracellular bulk by ~22 kJ/mol as compared to the extracellular side (black arrow). (**B**) Close-up views comparing the bound positions of pentamidine (left) and melarsoprol (right) and showing the mutated sites and major interactions with the AQP2 pore lining.

The online version of this article includes the following figure supplement(s) for figure 8:

**Figure supplement 1.** Backbone RMSD of the protein inserted into a lipid bilayer showing convergence to ~3 Å in a simulation of 100 ns length.

## The linker length and composition is a strong determinant for high affinity binding of pentamidine

We determined the $K_i$ values for analogues with a 2–8 methylene unit linker (*Figure 9A*, *Table 1*). Pentamidine analogues featuring 5–7 units displayed submicromolar binding affinities (5 > 6 > 7),

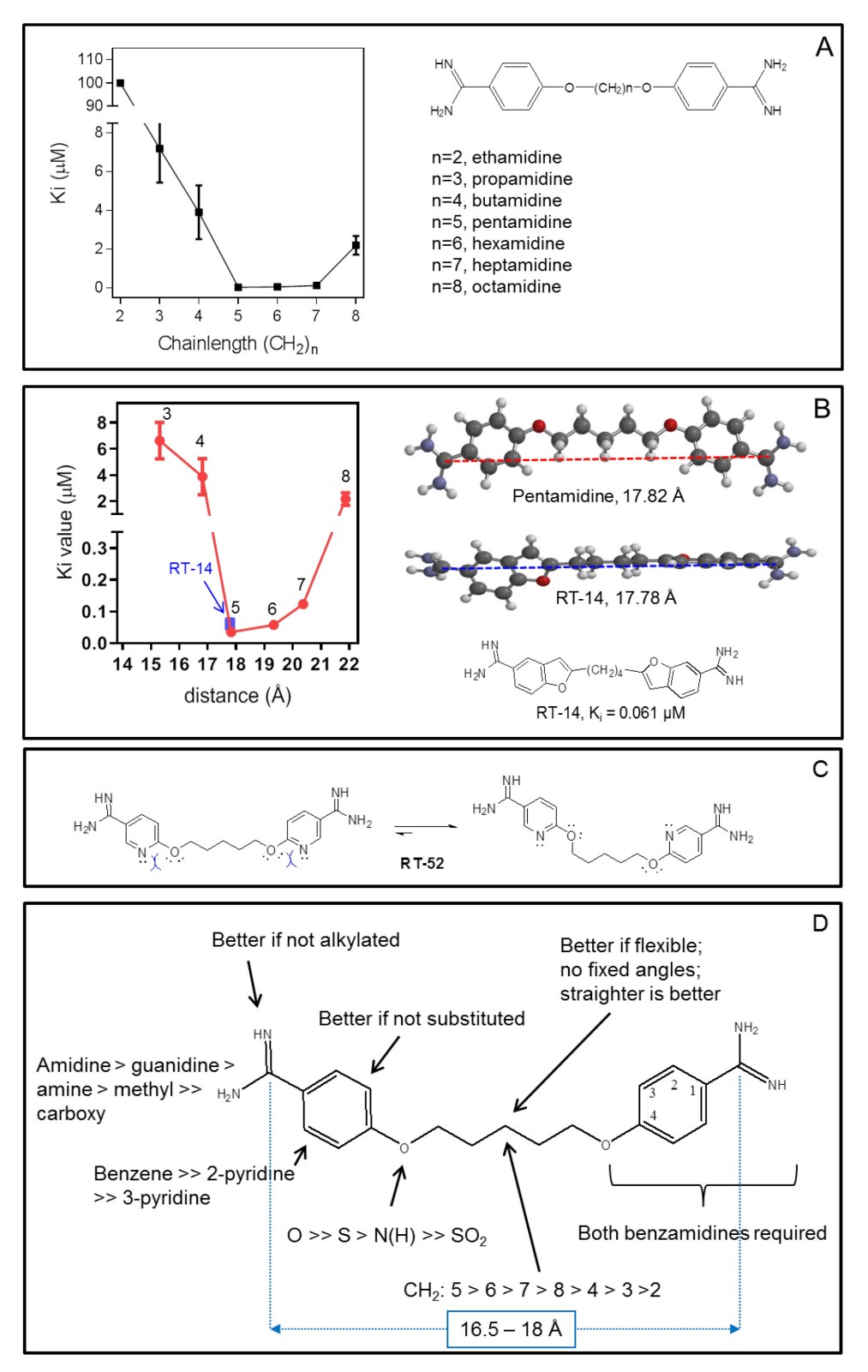

**Figure 9.** Correlation between linker chain length and affinity to HAPT1. (**A**) A series of pentamidine analogues with different methylene linker length was tested for inhibition of TbAQP2/HAPT1-mediated 25 nM [³H]-pentamidine transport (i.e. in the presence of adenosine to block the TbAT1/P2 transporter). The $K_i$ values are listed in *Table 1*. All $K_i$ values are shown as average and SEM of 3 or more independent experiments, each performed in triplicate. (**B**) The distance between the amidine carbon atoms in the lowest-energy conformation was calculated using density functional theory as

*Figure 9 continued on next page*

**eLife** Research article

Biochemistry and Chemical Biology | Microbiology and Infectious Disease

*Figure 9 continued*

implemented in Spartan' 16 v2.0.7. Geometry optimisations were performed with the wB97XD functional and the 6–31G* basis set at the ground state in gas phase. Structures and distances shown represent the dication state that is overwhelmingly prevalent in aqueous solution at neutral pH. The numbered red data points correspond to the propamidine - octamidine series in frame A. (**C**) Repulsion between free electron pairs (double dots), indicated by curved blue lines for RT-52 in the *cis*-conformation, causing it to exist overwhelmingly in the *anti*-conformation. (**D**) Overview of SAR observations on the binding preferences of TbAQP2 for pentamidine and its analogues. All data for these graphs are contained in *Figure 9—source data 1*.

The online version of this article includes the following source data and figure supplement(s) for figure 9:

**Source data 1.** $K_i$ values for *Figure 9A and B*.

**Figure supplement 1.** Correlation between the Resistance Factor (RF; $EC_{50}$(aqp2/3 null)/$EC_{50}$(TbAQP2-WT)) and the $K_i$ value for inhibition of the High Affinity Pentamidine Transporter (HAPT1) encoded by TbAQP2.

while fewer (3-4) or more (8) only conveyed low micromolar binding affinity, equivalent to a decrease in Gibbs free energy of binding ($\Delta G^0$) from 10.2 to 13.0 kJ/mol (*Table 1*). Energy minimalization using Gaussian16 yielded an elongated conformation for pentamidine, with an inter-amidine length of 17.8 Å (*Figure 7B*). Replacement of the ether oxygens with S or NH, analogues RT-48 and RT-50, respectively (*Table 1*) resulted in $\delta(\Delta G^0)$ of 10.0 and 12.9 kJ/mol, respectively, indicating that the ether oxygens potentially act as H-bond acceptors: the NH group serves only as an H-bond donor, as its lone pair is conjugated with the aromatic system, and the sulphur mimics an aromatic NH (*Beno et al., 2015*). The sulfone analogue (RT-49), which introduces a dihedral angle of 180° between the benzamidine and the linker (*Brameld et al., 2008*), displayed no binding affinity. We propose that a near-planar conformation of the Phe-O-CH$_2$ segment is required for efficient engagement of the binding site. This is supported by examining the binding affinities found for the analogous benzofuramidine series (e.g. RT-14, *Figure 9B*), which has a conformationally predefined ether-methylene bond orientation. Replacement of the middle methylene unit of pentamidine with an isosteric oxygen (ethylene glycol derivative DB1699, *Table 1*) results in a less flexible linker and a remarkable drop in binding affinity ($\delta(\Delta G^0)$=15.3 kJ/mol).

## Two amidine groups are required for high affinity binding

Matched-molecular pair analysis of non-symmetric analogues identified that both amidines contribute to high affinity binding (compare pairs pentamidine/RT-36 and pentamidine/CHI/1/72/1; *Table 1*). Removal of an H-bond donor (as in CHI/1/72/1) leads to a loss in $\Delta G^0 > 10$ kJ/mol. The aniline derivative RT-36 can still act as an H-bond donor, albeit with significantly reduced basicity (and thus H-bond acceptor propensity), and accordingly displayed intermediate affinity ($\delta(\Delta G^0)$=6.2 kJ/mol). Interestingly, the removal of one amidine (compare butamidine and CHI/1/69/1) did not produce a significant effect on the binding affinity ($K_i$ = 3.87 µM and $K_i$ = 2.33 µM, respectively), indicating that the low affinity of butamidine (compare 36 nM for pentamidine) is due to an inability to attain a productive interaction with the second amidine. Capping of the amidine group, resulting in imidazoline analogue RT-32, or methylation (analogue RT-30) reduced binding to HAPT1, probably due to increased steric crowding at the interaction site, impairing H-bonding. Reducing pentamidine to just 4-hydroxybenzamidine removed essentially all affinity ($K_i$ = 2.9 mM; $\delta(\Delta G^0)$=28.1 kJ/mol), and the replacement of one amidine with a carboxylic group (compare propamidine, RT-38) was highly deleterious for engagement with the binding site. Finally, the orientation of the amidine group is crucial as shown by a *meta* to *para* change on the phenyl ring (*meta*-pentamidine, *Table 1*). We conclude that for high affinity both amidine groups must be able to interact unimpeded with the transporter, and in the linear (*para*) conformation.

## Fully conjugated linking units

Stilbamidine and the short-linker analogues FR39 and CRMI8 (*Ríos Martínez et al., 2015*) displayed low binding affinity (*Table 1*). Diminazene also displayed similar low affinity ($K_i$ = 63 µM), and [$^3$H]-diminazene uptake can only just be detected in procyclic *T. b. brucei*, that is in the absence of the TbAT1/P2 transporter (*Teka et al., 2011*), potentially indicating a minimal uptake via HAPT1. Stilbamidine and diminazene feature a similar inter-amidine distance, much shorter than pentamidine (12.35 and 12.25 Å, respectively). DB75 (furamidine) likewise displayed low affinity (*Table 2*) and is

**Table 1.** Pentamidine analogues with an aliphatic linker.

| Compound | $R_1$ | $R_2$ | $R_3$ | $R_4$ | X | Y | Z | $K_i$ ($\mu$M) | $\delta(\Delta G^0)$ PMD (kJ/mol) |
|---|---|---|---|---|---|---|---|---|---|
| Ethamidine | Am | Am | H | H | CH | CH | $-O-(CH_2)_2-O-$ | >100 | >19.7 |
| Propamidine | Am | Am | H | H | CH | CH | $-O-(CH_2)_3-O-$ | 6.63 ± 1.40 | 13.0 |
| Butamidine | Am | Am | H | H | CH | CH | $-O-(CH_2)_4-O-$ | 3.87 ± 1.38 | 11.7 |
| Pentamidine (PMD) | Am | Am | H | H | CH | CH | $-O-(CH_2)_5-O-$ | **0.036 ± 0.0006** | – |
| Hexamidine | Am | Am | H | H | CH | CH | $-O-(CH_2)_6-O-$ | 0.058 ± 0.011 | 1.3 |
| Heptamidine | Am | Am | H | H | CH | CH | $-O-(CH_2)_7-O-$ | 0.123 ± 0.010 | 3.1 |
| Octamidine | Am | Am | H | H | CH | CH | $-O-(CH_2)_8-O-$ | 2.16 ± 0.48 | 10.2 |
| RT-48 | Am | Am | H | H | CH | CH | $-S-(CH_2)_5-S-$ | 2.01 ± 0.86 | 10.0 |
| RT-50 | Am | Am | H | H | CH | CH | $-NH-(CH_2)_5-NH-$ | 6.27 ± 1.30 | 12.9 |
| RT-49 | Am | Am | H | H | CH | CH | $-SO_2-(CH_2)_5-SO_2-$ | >150 | >20.7 |
| DB1699 | Am | Am | H | H | CH | CH | $-O-(CH_2)_2-O-(CH_2)_2-O-$ | 16.6 ± 2.1 | 15.3 |
| RT-36 | Am | $NH_2$ | H | H | CH | CH | $-O-(CH_2)_5-O-$ | 0.43 ± 0.07 | 6.2 |
| CHI/1/72/1 | Am | $CH_3$ | H | H | CH | CH | $-O-(CH_2)_5-O-$ | 3.1 ± 0.7 | 10.7 |
| CHI/1/69/1 | Am | H | H | H | CH | CH | $-O-(CH_2)_4-O-$ | 2.3 ± 0.5 | 10.4 |
| RT-38 | Am | CA | H | H | CH | CH | $-O-(CH_2)_3-O-$ | NI, 100 | >19.7 |
| *meta*-PMD | H | H | Am | H | CH | CH | $-O-(CH_2)_5-O-$ | 2890 ± 1050 | 28.1 |
| RT-32 | Im | Im | H | H | CH | CH | $-O-(CH_2)_5-O-$ | 0.40 ± 0.08 | 6.0 |
| RT-30 | MeAm | MeAm | H | H | CH | CH | $-O-(CH_2)_5-O-$ | 0.30 ± 0.07 | 5.3 |
| Stilbamidine | Am | Am | H | H | CH | CH | $-CH = CH-$ | 54.8 ± 3.2 | 18.3 |
| FR39 | G1 | G1 | H | H | CH | CH | $-(CH_2)_2-$ | 41.7 ± 15.2 | 17.6 |
| CRMI8 | G2 | G2 | H | H | CH | CH | $-(CH_2)_2-$ | 52.8 ± 12.7 | 18.1 |
| RT-43 | Am | Am | H | Cl | CH | CH | $-O-(CH_2)_5-O-$ | 0.51 ± 0.15 | 6.6 |
| Iodo-PMD | Am | Am | H | I | CH | CH | $-O-(CH_2)_5-O-$ | 2.15 ± 0.04 | 8.4 |
| RT-46 | Am | Am | H | $-C(O)NH_2$ | CH | CH | $-O-(CH_2)_5-O-$ | >100 | >19.7 |
| RT-52 | Am | Am | H | H | CH | N | $-O-(CH_2)_5-O-$ | 8.84 ± 0.88 | 13.7 |
| RT-53 | Am | Am | H | H | N | CH | $-O-(CH_2)_5-O-$ | NI, 250 | >22 |

Am, amidine; MeAm, Methyl-amidine; Im, imidazole; CA, carboxylic acid; G1, 2-aminoimidazoline; G2, 1-methoxy-2-aminoimidazoline. PMD, pentamidine, NI, no inhibition at the indicated concentration in $\mu$M. $K_i$ is the inhibition constant for [$^3$H]-pentamidine transport by TbAQP2/HAPT1. $\delta(\Delta G^0)$ PMD is the difference in Gibbs Free Energy of interaction of the substrate with TbAQP2 with the same value for pentamidine (PMD). All $K_i$ values are the average and SEM of at least 3–4 independent experiments.

only internalised by TbAT1/P2 (*Ward et al., 2011*). The 2,5-furan linker imposes a fixed, inflexible angle of 136° on the benzamidine moieties and the phenyl rings will adopt a planar orientation with respect to the furan plane. This appears to allow only one benzamidine end to interact with the transporter, as (unlike the flexible linker of aliphatic diamidines, *vide supra*) the replacement of one

**Table 2.** Selection of diamidine analogues with aromatic linkers.

| Compound | $R_1$ | $R_2$ | Ar | X | Y | $K_i$ (µM) | $\delta(\Delta G^0)$ PMD (kJ/mol) |
|---|---|---|---|---|---|---|---|
| DB75 | Am | Am | | CH | CH | 38.2 ± 10.2 | 17.3 |
| DB607 | Am | OCH₃ | | CH | CH | 18.1 ± 1.9 | 15.5 |
| DB960 | Am | NMB[a] | | CH | CH | 16.6 ± 3.5 | 15.3 |
| DB994 | Am | Am | | N | CH | 167 ± 20 | 21.0 |
| DB829 | Am | Am | | CH | N | 39.9 ± 8.0 | 17.4 |
| DB1061 | EtAm | EtAm | | CH | CH | 32.3 ± 6.0 | 16.9 |
| DB1062 | 2MeIm | 2MeIm | | CH | CH | 59.6 ± 11.2 | 18.4 |
| ER1004 | Am | Am | | CH | CH | 68.7 ± 16.0 | 18.8 |
| DB320 | Am | Am | | CH | CH | 71.3 ± 12.1 | 18.9 |
| DB686 | Gua | Gua | | CH | CH | 0.29 ± 0.11 | 5.2 |
| DB1063 | EtAm | EtAm | | CH | CH | 0.40 ± 0.10 | 6.0 |
| DB1064 | 2MeIm | 2MeIm | | CH | CH | 3.0 ± 0.82 | 11.0 |
| DB1213 | Am | Am | | CH | CH | 0.72 ± 0.17 | 7.5 |
| DB1077 | Am | Am | | CH | CH | 13.8 ± 3.1 | 14.8 |
| DB914 | Am | Am | | CH | CH | 0.073 ± 0.013 | 1.8 |

Am, amidine; Im, imidazole; EtAm, ethylamidine; 2MeIm, 2-methylimidazoline; NMB, *N*-methyl benzimidazole. [a]This compound lacks the second benzene ring and features the terminal NMB moiety instead. $K_i$ is the inhibition constant for [³H]-pentamidine transport by TbAQP2/HAPT1. $\delta(\Delta G^0)$ PMD is the difference in Gibbs Free Energy of interaction of the substrate with TbAQP2 with the same value for pentamidine (PMD). All $K_i$ values are the average and SEM of at least 3–4 independent experiments.

amidine group actually increases the binding affinity, presumably by allowing an improved bonding orientation of the remaining amidine. Thus, DB607 (methoxy for amidine) and DB960 (*N*-methyl benzimidazole for benzamidine) display a somewhat higher affinity than DB75, although the fixed angle was unchanged. Introduction of a pyridine-N in the *ortho*-position with respect to the amidine functionality (DB994), dramatically reduces the $pK_a$ of the amidine moiety (*Wang et al., 2010*), resulting in a complete loss of binding affinity ($K_i = 167 \pm 20$ µM), while this was not observed for the corresponding *meta*-pyridine derivative (DB829). The unfavourable furan bond angle is further demonstrated by the distally elongated analogues DB1061 and DB1062 that approximate the inter-amidine distance of pentamidine but showed no improvement in binding affinity (*Table 2*). Replacement of furan with thiazole (ER1004) or methylpyrrole (DB320), which feature a similar bond angle, also revealed comparable binding affinities. In contrast, a 2,5-substituted thiophene (DB686 and DB1063) or 2,5-substituted selenophene (DB1213) as a bio-isosteric replacement for the furan linker resulted in significantly higher binding affinities when compared to their matched pair analogue (DB1063/DB1061 and DB1213/DB75), which we attribute to a larger benzamidine-benzamidine angle. This is corroborated by the much weaker binding of the 2,4-thiophene derivative DB1077. A terminal amidine cap (imidazoline) reduced affinity as it did for pentamidine (compare DB1061/ DB1062 and DB1063/DB1064 (*Table 2*)). A difuran spacer (DB914) resulted in a high affinity binder ($K_i = 0.073$ µM) because the two furans orient themselves in a *trans* conformation, resulting in a near-linear molecule.

## Modifications to the phenyl rings of pentamidine

Substituents in the *ortho*-position (relative to the alkyloxy substituent) of pentamidine were poorly tolerated, including chloride or iodide (RT-43, iodopentamidine; *Table 1*); the amide analogue displayed no affinity at all (RT-46). Such substituents will cause an out-of-plane conformation of the alkoxy-group to avoid clashing with the *ortho*-substituent; high-affinity pentamidine binding appears to require a coplanar arrangement of the first methylene bound to the oxygen. Similarly, the introduction of an *ortho*-pyridine N (RT-52) led to a $\delta(\Delta G^0)$ of 13.7 kJ/mol. This derivative exhibits a conformational bias towards an *anti*-orientation of the ether oxygen and pyridine nitrogen (*Figure 9C*; *Chein and Corey, 2010*). The regio-isomeric *meta*-pyridine (RT-53) was completely inactive, reflecting the need for a positively charged amidine, as this analogue has a significantly reduced pKa (*Wang et al., 2010*) (see furan-spaced analogue DB994, *supra*).

## Non-diamidine trypanocides

The important veterinary trypanocide isometamidium, a hybrid of the phenanthridine ethidium and the diamidine diminazene, inhibited HAPT1-mediated [³H]-pentamidine uptake with a $K_i$ of only 3.5 µM (*Supplementary file 1*), most probably through an interaction with its benzamidine moiety, as ethidium displayed virtually no affinity ($K_i = 97$ µM). However, we found no evidence that HAPT1/ AQP2 is able to transport the bulky isometamidium molecule. For instance, the 2T1, *tbaqp2* null, TbAQP2 expressed in *tbaqp2* null, and the *tbaqp2/tbaqp3* null strains displayed statistically identical $EC_{50}$ values for isometamidium ($112 \pm 12$ nM, $103 \pm 14$ nM, $98 \pm 24$ nM and $95 \pm 12$ nM, respectively; $p > 0.05$, Student's unpaired t-test), and the $EC_{50}$ values for ethidium were also identical for each of these strains ($1.32 \pm 0.07$ µM, $1.39 \pm 0.08$ µM, $1.35 \pm 0.11$ µM and $1.38 \pm 0.14$ µM, respectively). It is thus likely that isometamidium acts as an extracellular inhibitor rather than a substrate for HAPT/AQP2, as it does for the TbAT1/P2 transporter (*de Koning, 2001b*). The nitro-heterocyclic trypanocide megazol (*Carvalho et al., 2014*), curcumin and its trypanocidal analogue AS-HK14 (*Alkhaldi et al., 2015*) failed to inhibit HAPT1. Two trypanocidal bis-phosphonium compounds, CD38 (*Taladriz et al., 2012*) and AHI43 (*Alkhaldi et al., 2016*) did inhibit pentamidine uptake ($K_i$ 5– 10 µM), whereas two related compounds, CDIV31 and AHI15 (*Taladriz et al., 2012*), did not. Phloretin, which inhibits human AQP9 and AQP3 (*Geng et al., 2017*),[47] displayed a $K_i$ of 1.76 µM for HAPT1/TbAQP2.

## Are all the HAPT1/AQP2 inhibitors transported?

In an uptake-by-endocytosis model some correlation between TbAQP2 binding energy and TbAQP2-mediated uptake rates for each analogue would be expected, although in many instances of receptor-mediated endocytosis factors such as gating and the induction of conformational

changes in the carrier might complicate such a correlation. Nevertheless, if we could observe a significant correlation of these parameters it would strengthen the argument for the endocytosis model. We were unable to ascertain the existence of such a correlation directly, for lack of radiolabelled substrates other than pentamidine and diminazene and thus used the Resistance Factor (RF; $EC_{50}$(aqp2/3 null)/$EC_{50}$(TbAQP2-WT)) as a proxy: clearly, a compound with a significant RF is internalized by TbAQP2. We observed a poor correlation between HAPT1 binding affinity and the level of resistance in the *tbaqp2/tbaqp3* null strain ($r^2$ = 0.039, *Figure 9—figure supplement 1*; n = 30), with many inhibitors, even those with high affinity, not displaying any significant resistance in the null line. This indicates that many of these compounds inhibit HAPT1/TbAQP2 but are not transported by it. This fails to support a model in which pentamidine binds and is then internalized by endocytosis: the inhibitors do not show resistance in the *tbaqp2/tbaqp3* null line, whereas substrates do. The lack of any correlation would be especially problematic for a model of 'passive' pentamidine endocytosis, where the drug merely piggy-backs on TbAQP2 as it is internalised in its regular turnover schedule ($t_{1/2}$ >4 h [*Quintana et al., 2020*]), without inducing any conformation changes in the receptor protein. The caveat inherent to using the RF instead of rate of transport is that it cannot be excluded that some of the test compounds are AQP2 substrates yet predominantly taken up by transporters other than TbAQP2, and hence show a low RF.

## SAR summary

*Figure 9D* summarises the structure-activity relationship of pentamidine interactions with HAPT1/TbAQP2. No modification in any part of pentamidine improved affinity for TbAQP2, but virtually every modification resulted in a significant loss of binding activity (a similar analysis with melaminophenyl arsenicals was impossible for lack of the required organo-arsenicals). The results clearly demonstrate that at least both amidine groups and one or both ether oxygens are involved in interactions with AQP2, the sum of which adds up to the unusually high binding energy for this substrate-transporter pair ($\Delta G^0$ = −42.6 kJ/mol). These results are fully compatible with pentamidine binding in an elongated orientation, and are in complete agreement with the modelling and molecular dynamics, and the mutational analysis presented above, strengthening those conclusions using a completely different approach.

## Discussion

There is overwhelming consensus that expression of TbAQP2 is associated with the extraordinary sensitivity of *T. brucei* to pentamidine and melaminophenyl arsenicals, and that mutations and deletions in this locus cause resistance (*Baker et al., 2012*, *Baker et al., 2013*; *Graf et al., 2013*; *Graf et al., 2015*; *Graf et al., 2016*; *Pyana Pati et al., 2014*; *Munday et al., 2014*; *Munday et al., 2015a*; *Unciti-Broceta et al., 2015*). What has remained unclear, however, is the mechanism underpinning these phenomena – there are currently no documented other examples of aquaporins transporting such large molecules. Yet, considering how ubiquitous aquaporins are to almost all cell types, this question is of wide pharmacological importance: if large cationic and neutral drugs (pentamidine and melarsoprol, respectively) can be taken up via an aquaglyceroporin of *T. brucei*, what other pharmacological or toxicological roles may these channels be capable of in other cell types? This manuscript clearly shows that changes in the TbAQP2 WGYR and NPA/NPA motifs, which collectively enlarge the pore and remove the cation filter, allow the passage of these drugs into the cell, and thereby underpin the very high sensitivity of the parasite to these drugs.

TbAQP2 has evolved, apparently by positive selection given the high dN/dS ratio, to remove all main constriction points, including the aromatic amino acids and the cationic arginine of the ar/R selectivity filter, and the NPA/NPA motif, resulting in an unprecedentedly enlarged pore size (*Baker et al., 2012*; *Munday et al., 2015a*). Whereas the advantage of this to *T. b. brucei* is yet unknown, the adaptation is stable within the *brucei* group of trypanosomes, and found in *T. b. rhodesiense* (*Munday et al., 2014*; *Graf et al., 2016*), *T. b. gambiense* (*Graf et al., 2013*; *Graf et al., 2015*; *Munday et al., 2014*; *Pyana Pati et al., 2014*), *T. equiperdum* and *T. evansi* (Philippe Büscher and Nick Van Reet, unpublished). As such, it is not inappropriate to speculate that the wider pore of TbAQP2 (i) allows the passage of something not transported by TbAQP1 and TbAQP3; (ii) that this confers an a yet unknown advantage to the cell; and (iii) that uptake of pentamidine is a by-product of this adaptation.

It is difficult to reconcile the literature on pentamidine transport/resistance with uptake via endocytosis. For instance, the rate of endocytosis in bloodstream trypanosomes is much higher than in the procyclic lifecycle forms (*Langreth and Balber, 1975*; *Zoltner et al., 2016*), yet the rate of HAPT-mediated [³H]-pentamidine uptake in procyclics is ~10 fold higher than in bloodstream forms (*de Koning, 2001a*; *Teka et al., 2011*), despite the level of TbAQP2 expression being similar in both cases (*Siegel et al., 2010*; *Jensen et al., 2014*). Moreover, in procyclic cells TbAQP2 is spread out over the cell surface (*Baker et al., 2012*) but endocytosis happens exclusively in the flagellar pocket (*Field and Carrington, 2009*) (which is 3-fold smaller in procyclic than in bloodstream forms [*Demmel et al., 2014*]), as the pellicular microtubule networks below the plasma membrane prevent endocytosis (*Zoltner et al., 2016*). Thus, TbAQP2-mediated pentamidine uptake should be all but impossible in procyclic *T. brucei*, if dependent on endocytosis. Similarly, the expression of TbAQP2 in *Leishmania mexicana* promastigotes produced a rate of [³H]-pentamidine uptake more than 10-fold higher than observed in *T. brucei* BSF (*Munday et al., 2014*), despite these cells also having a low endocytosis rate (*Langreth and Balber, 1975*). The $K_m$ and inhibitor profile of the TbAQP2-mediated pentamidine transport in these promastigotes was indistinguishable from HAPT in procyclic or bloodstream form *T. brucei* (*de Koning, 2001a*).

The experimental $V_{max}$ for HAPT-mediated pentamidine uptake in *T. brucei* BSF and procyclics (*de Koning, 2001a*) can be expressed as $9.5 \times 10^5$ and $8.5 \times 10^6$ molecules/cell/h, respectively; given a 1:1 stoichiometry for AQP2:pentamidine the endocytosis model would require the internalisation and recycling of as many units of TbAQP2 and this seems unlikely, especially in procyclic cells, as even in BSF the half-life time for TbAQP2 turnover is >4 hr (*Quintana et al., 2020*) and procyclic cells have a lower endocytosis rate and cannot easily internalise the aquaporins spread over the cell surface, as discussed above. Given the observed rate of uptake and turnover rate, this would require the presence of $\sim 4 \times 10^6$ TbAQP2 units per BSF cell in the flagellar pocket. These observations are all inconsistent with the contention that pentamidine uptake by trypanosomes is principally dependent on endocytosis. Although it is likely that AQP2-bound pentamidine is internalised as part of the natural turnover rate of the protein, this is not likely to contribute very significantly to the overall rate of uptake of this drug.

Furthermore, the Gibbs free energy of −42 kJ/mol for the pentamidine/AQP2 interaction (*de Koning, 2001a*; *Zoltner et al., 2016*) is highly unlikely to be the result of the one interaction between one terminal amidine and Asp265 as required in the endocytosis model (*Song et al., 2016*). For the TbAT1 transporter, a double H-bond interaction of Asp140 with the N1(H)/C(6)NH₂ motif of adenosine or with one amidine of pentamidine (*Munday et al., 2015b*) is estimated to contribute only ~16 kJ/mol to the total $\Delta G^0$ of −34.5 kJ/mol for adenosine (−36.7 kJ/mol, pentamidine) (*de Koning and Jarvis, 1999*). The endocytosis model also does not address the internalisation of melaminophenyl arsenicals, which presumably would equally need access to Asp265, or address why most diamidines including furamidines and diminazene aceturate are at best extremely poor substrates for TbAQP2 (*Teka et al., 2011*; *Ward et al., 2011*).

Here we systematically mapped the interactions between the aquaporin and pentamidine ($\Delta G^0$ for 71 compounds), yielding a completely consistent SAR with multiple substrate-transporter interactions, summarised in *Figure 9D*. The evidence strongly supports the notion that pentamidine engages TbAQP2 with both benzamidine groups and most probably with at least one of the linker oxygens, and that its flexibility and small width are both required to optimally interact with the protein. This is completely corroborated by molecular dynamics modelling, which shows minimal energy to be associated with a near-elongated pentamidine centrally in the TbAQP2 pore, without major energy barriers to exiting in either direction, but driven to the cytoplasmic side by the membrane potential. This contrasts with the contention (*Song et al., 2016*) that pentamidine could not be a permeant for TbAQP2 because it did not transport some small cations and that this proves that the larger pentamidine cannot be a substrate either. There is scant rationale for that assertion: out of many possible examples: there are 5 orders of magnitude difference in affinity for pentamidine and *para*-hydroxybenzamidine (35 nM *vs* 2.9 mM; *Supplementary file 1*); adenine is not a substrate for the *T. brucei* P1 adenosine transporter (*de Koning and Jarvis, 1999*), the SLC1A4 and SLC1A5 neutral amino acid transporters transport Ala, Ser, Cys and Thr but not Gly (*Kanai et al., 2013*), Na⁺ is not a permeant of K⁺ channels (*Zhorov and Tikhonov, 2013*) and some NAT family transporters from bacterial, plant and fungal species display much higher affinity for xanthine and uric acid but not for hypoxanthine (*Gournas et al., 2008*).

The endocytosis model identifies only two key residues for pentamidine access (Leu264) and binding (Asp265) in TbAQP2 (*Song et al., 2016*). Yet, multiple clinical isolates and laboratory strains contain chimeric AQP2/3 genes associated with resistance and/or non-cure that have retained those residues and should thus allow binding and internalisation of pentamidine (*Graf et al., 2013*; *Pyana Pati et al., 2014*; *Unciti-Broceta et al., 2015*; *Munday et al., 2014*). Although we find that introduction of the AQP3 Arg residue in position 264 (TbAQP2$^{L264R}$) disables pentamidine transport, we would argue that this is because the positively charged arginine, in the middle of the pore, is blocking the traversing of all cations through the pore, as is its common function in aquaporins (*Beitz et al., 2006*; *Wu et al., 2009*). Indeed, the W(G)YR filter residues appear to be key determinants for pentamidine transport by AQPs and the introduction of all three TbAQP2 residues into TbAQP3 (AQP3$^{W102I/R256L/Y250L}$) was required to create an AQP3 that at least mildly sensitised to pentamidine, and facilitated a detectable level of pentamidine uptake. Conversely, any one of the mutations I110W, L258Y or L264R was sufficient to all but abolish pentamidine transport by TbAQP2. Similarly, the conserved NPA/NPA motif, and particularly the Asp residues, present in TbAQP3 but N<u>S</u>A/NP<u>S</u> in TbAQP2, is also associated with blocking the passage of cations (*Wree et al., 2011*). The unique serine residues in this TbAQP2 motif, halfway down the pore, might be able to make hydrogen bonds with pentamidine. Reinstating the NPA/NPA motif resulted in a TbAQP2 variant with a 93.5% reduced rate of [$^3$H]-pentamidine transport.

Tryptophan residues were introduced towards the cytoplasmic end of the TbAQP2 pore (L84W, L118W, L218W) to test the hypothesis that introducing bulky amino acids in that position would block the passage of pentamidine. Each of these mutants was associated with reduced sensitivity to pentamidine and cymelarsan and a > 90% reduction in [$^3$H]-pentamidine uptake. This effect was size-dependent as the pentamidine transport rate of L84M and L218M was statistically identical to that of control TbAQP2 cells, and L118M also displayed a higher transport rate than L118W (p<0.0001). These mutant AQPs were still functional aquaglyceroporins as their expression in *tbaqp1-2-3* null cells made those cells less sensitive to the TAO inhibitor SHAM, and increased glycerol uptake.

Independence from endocytosis was investigated by employing the tetracycline-inducible CRK12 RNAi cell line previously described to give a highly reproducible and progressive endocytosis defect in *T. brucei* (*Monnerat et al., 2013*), with the aim to distinguish between uptake via endocytosis and transporters, as current evidence suggests that in *T. brucei* all endocytosis, taking place exclusively in the flagellar pocket, is clathrin-dependent and AP-2 independent (*Morgan et al., 2002*; *Allen et al., 2003*). This means that the endocytotic mechanisms of TbAQP2 and suramin receptor ISG75, which are both directed to the lysosome after ubiquitylation (*Quintana et al., 2020*; *Zoltner et al., 2015*), are likely to be similar enough for a direct comparison. Twelve hours after CRK12 RNAi induction pentamidine transport was not significantly reduced although uptake of [$^3$H]-suramin, which is accumulated by endocytosis through the *T. brucei* flagellar pocket (*Zoltner et al., 2016*), was reduced by 33% (p=0.0027), indicating successful timing of the experiment to the early stage of endocytosis slow-down. We also show that pentamidine, at approximately its half-maximal occupancy concentration, did not influence the half-life time of TbAQP2 turnover, as is often the case in receptor-mediated endocytosis. Nor should TbAQP2, which exists as a rigid tetramer of tetramers in *T. brucei* bloodstream forms (*Quintana et al., 2020*), be able to undergo the type of conformational change necessary to signal receptor occupancy and internalisation as observed in well-documented examples of ligand-triggered internalisation of transporters (e.g. *Gournas et al., 2010*; *Gournas et al., 2017*; *Ghaddar et al., 2014*; *Keener and Babst, 2013*). Thus, the combined evidence, taken together, strongly suggests that pentamidine is not taken up by endocytosis, not induce endocytosis of TbAQP2.

Although several ionophores, including CCCP, nigericin and gramicidin strongly inhibited pentamidine uptake, similar to what has been previously reported for transport processes in *T. brucei* that are linked to the protonmotive force (*de Koning and Jarvis, 1997a*; *de Koning and Jarvis, 1997b*; *de Koning and Jarvis, 1998*; *de Koning et al., 1998*), this is probably due to the inside-negative membrane potential of −125 mV (*de Koning and Jarvis, 1997b*) attracting the dicationic pentamidine. This is consistent with the prediction of the molecular dynamics modelling, and the reported role of the HA1–three proton pumps in pentamidine but not melarsoprol resistance (*Alsford et al., 2012*; *Baker et al., 2013*). Although CCCP does inhibit pentamidine through direct, competitive inhibition of TbAQP2 as well, this only starts to have a measurable impact above ~5 μM (IC$_{50}$ of 20.7 μM), whereas its effects after preincubation, i.e. the combination of competitive

inhibition and reducing the protonmotive force, shows ~63% inhibition of pentamidine transport at 1 µM and ~90% inhibition at 5 µM, showing that the more important effect of CCCP is via reduction of the PMF. This is consistent with the conclusion from the molecular dynamics analysis that inward pentamidine flux is dependent on the inside-negative membrane potential.

Altogether, we conclude that the primary entry of the sleeping sickness drugs pentamidine and melarsoprol into *T. brucei* spp. is through the unusually large pore of TbAQP2, rendering the parasite extraordinarily sensitive to the drugs (compare *Leishmania mexicana* [*Munday et al., 2014*]). This is the first report providing detailed mechanistic evidence of the uptake of organic drugs (of MW 340 and 398, respectively) by an aquaporin. We show that this porin has evolved through positive selection and identify the adaptations in the constriction motifs that enabled it. We consider that other pore-opening adaptations may have evolved in other organisms, including pathogens, which could initiate the pharmacological exploitation of aquaporins and lead to the design of new drug delivery strategies.

# Materials and methods

## Key resources table

| Reagent type (species) or resource | Designation | Source or reference | Identifiers | Additional information |
|---|---|---|---|---|
| Cell line (*Trypanosoma brucei*) | 2T1 | David Horn | | |
| Cell line (*Trypanosoma brucei*) | *aqp2/aqp3* null | David Horn | | |
| Cell line (*Trypanosoma brucei*) | *aqp1-3* null | David Horn | | |
| Cell line (*Trypanosoma brucei*) | *CRK12 RNAi* | Tansy Hammarton | | |
| Recombinant DNA reagent | pRPa | David Horn | | plasmid for expression in *T. brucei* |
| Gene (*Trypanosoma brucei*) | AQP2 | TriTrypDB | Tb927.10.14170 | Sequence in **Supplementary file 2** |
| Gene (*Trypanosoma brucei*) | TbAQP2$^{S131P/S263A}$ | This paper | | Mutated TbAQP2; sequence in **Supplementary file 2** |
| Gene (*Trypanosoma brucei*) | TbAQP2$^{L258Y}$ | This paper | | Mutated TbAQP2; sequence in **Supplementary file 2** |
| Gene (*Trypanosoma brucei*) | TbAQP2$^{I110W}$ | This paper | | Mutated TbAQP2; sequence in **Supplementary file 2** |
| Gene (*Trypanosoma brucei*) | TbAQP2$^{L264R}$ | This paper | | Mutated TbAQP2; sequence in **Supplementary file 2** |
| Gene (*Trypanosoma brucei*) | TbAQP2$^{I110W\ /L264R}$ | This paper | | Mutated TbAQP2; sequence in **Supplementary file 2** |
| Gene (*Trypanosoma brucei*) | TbAQP2$^{I190T}$ | This paper | | Mutated TbAQP2; sequence in **Supplementary file 2** |
| Gene (*Trypanosoma brucei*) | TbAQP2$^{W192G}$ | This paper | | Mutated TbAQP2; sequence in **Supplementary file 2** |
| Gene (*Trypanosoma brucei*) | TbAQP2$^{I190T/W192G}$ | This paper | | Mutated TbAQP2; sequence in **Supplementary file 2** |

*Continued on next page*

*Continued*

| Reagent type (species) or resource | Designation | Source or reference | Identifiers | Additional information |
|---|---|---|---|---|
| Gene (*Trypanosoma brucei*) | TbAQP2$^{L84W}$ | This paper | | Mutated TbAQP2; sequence in *Supplementary file 2* |
| Gene (*Trypanosoma brucei*) | TbAQP2$^{L84M}$ | This paper | | Mutated TbAQP2; sequence in *Supplementary file 2* |
| Gene (*Trypanosoma brucei*) | TbAQP2$^{L118W}$ | This paper | | Mutated TbAQP2; sequence in *Supplementary file 2* |
| Gene (*Trypanosoma brucei*) | TbAQP2$^{L118M}$ | This paper | | Mutated TbAQP2; sequence in *Supplementary file 2* |
| Gene (*Trypanosoma brucei*) | TbAQP2$^{L218W}$ | This paper | | Mutated TbAQP2; sequence in *Supplementary file 2* |
| Gene (*Trypanosoma brucei*) | TbAQP2$^{L218M}$ | This paper | | Mutated TbAQP2; sequence in *Supplementary file 2* |
| Gene (*Trypanosoma brucei*) | TbAQP2$^{L84W/L118W}$ | This paper | | Mutated TbAQP2; sequence in *Supplementary file 2* |
| Gene (*Trypanosoma brucei*) | AQP3 | TriTrypDB | Tb927.10.14160 | Sequence in *Supplementary file 2* |
| Gene (*Trypanosoma brucei*) | TbAQP3$^{W102I/R256L}$ | This paper | | Sequence in *Supplementary file 2* |
| Gene (*Trypanosoma brucei*) | TbAQP3$^{W102I/R256L/Y250L}$ | This paper | | Sequence in *Supplementary file 2* |
| Chemical compound, drug | | All itemised in *Supplementary file 1* | | Custom inhibitors and pentamidine analogues |
| Chemical compound, drug | Pentamidine isethionate | Sigma-Aldrich | | |
| Chemical compound, drug | Diminazene aceturate | Sigma-Aldrich | | |
| Chemical compound, drug | Suramin | Sigma-Aldrich | | |
| Chemical compound, drug | Cymelarsan | gift from C. Michael Turner | | |
| Chemical compound, drug | [$^3$H]-pentamidine | GE Healthcare Life Sciences | | Radiochemical |
| Chemical compound, drug | [$^3$H]-Suramin | American Radiolabeled Chemicals | | Radiochemical |
| Chemical compound, drug | [$^3$H]-glycerol | American Radiolabeled Chemicals | | Radiochemical |

## Trypanosome strains and cultures

The drug-sensitive clonal *T. b. brucei* strain 427 (MiTat 1.2/BS221) (*de Koning et al., 2000*) was used for all the work on the SAR of pentamidine transport. The *tbaqp2/tbaqp3* null cells (*Baker et al., 2012*) and *tbaqp1-2-3* null cells (*Jeacock et al., 2017*) (both obtained from David Horn, University of Dundee, UK) are derived from the 2T1 strain of *T. b. brucei* (*Alsford and Horn, 2008*). The *CRK12* RNAi cell line[28] was obtained from Dr Tansy Hammarton (University of Glasgow, UK) and is also based on the 2T1 cell line; RNAi expression was induced with 1 µg/ml tetracycline in the medium. All experiments were performed with bloodstream form trypanosomes grown in vitro in HMI-11 medium as described (*Wallace et al., 2002*) at 37°C in a 5% $CO_2$ atmosphere. Cultures

were routinely maintained in 10 ml of this medium, being seeded at $5 \times 10^4$ cells/ml and passed to fresh medium at reaching approximately $3 \times 10^6$ cells/ml after 48 hr. For transport experiments 150 or 200 ml of culture was seeded at the same density in large flasks and incubated until the culture reached late-log phase.

## Materials

A complete list of diamidine analogues and other chemicals used for the SAR study is given as a table with their sources (*Supplementary file 1*). Ionophores and uncouplers nigericin, gramicidin, carbonyl cyanide m-chlorophenyl hydrazone (CCCP) and valinomycin, as well as the *T. brucei* proton pump inhibitor *N*-ethylmaleimide (NEM) were all purchased from Sigma-Aldrich. New compounds synthesised for this study are listed and described in *Supplementary file 3*.

*Transport assays* - Transport of [$^3$H]-pentamidine was performed exactly as previously described for various permeants (*Wallace et al., 2002*; *Bridges et al., 2007*; *Teka et al., 2011*) in a defined assay buffer (AB; 33 mM HEPES, 98 mM NaCl, 4.6 mM KCl, 0.55 mM CaCl$_2$, 0.07 mM MgSO$_4$, 5.8 mM NaH$_2$PO$_4$, 0.3 mM MgCl$_2$, 23 mM NaHCO$_3$, 14 mM glucose, pH 7.3). [$^3$H]-pentamidine was custom-made by GE Healthcare Life Sciences (Cardiff, UK) with a specific activity of 88 Ci/mmol. Incubations of bloodstream form trypanosomes with 30 nM of this label (unless otherwise indicated) were performed in AB at room temperature for 60 s (unless otherwise indicated) and terminated by addition of 1 ml ice-cold 'stop' solution (1 mM unlabelled pentamidine (Sigma) in AB) and immediate centrifugation through oil (7:1 dibutylphthalate:mineral oil v/v (both from Sigma)). Transport was assessed in the presence of 1 mM adenosine to block uptake through the P2 aminopurine transporter; adenosine does not affect HAPT1-mediated transport (*de Koning, 2001a*; *Bridges et al., 2007*). Inhibition assays were performed routinely with 6–10 different concentrations of inhibitor over the relevant range, diluting stepwise by one third each time, in order to obtain a well-defined and accurate sigmoid plot and IC$_{50}$ value (inhibitor concentration giving 50% inhibition of pentamidine transport; calculated by non-linear regression using Prism 6.0 (GraphPad), using the equation for a sigmoid curve with variable slope). Highest concentration was usually 1 mM unless this was shown to be insufficient for good inhibition, or when limited by solubility. $K_i$ values were obtained from IC$_{50}$ values using

$$k_i = \mathrm{IC}_{50}/[1 + (\mathrm{L} + k_m)] \tag{1}$$

in which L is the [$^3$H]-pentamidine concentration and $K_m$ the Michaelis-Menten constant for pentamidine uptake by HAPT1 (*Wallace et al., 2002*). The Gibbs Free energy of interaction $\Delta G^0$ was calculated from

$$\Delta G^0 = -\mathrm{RT}\ln K_i \tag{2}$$

in which R is the gas constant and T is the absolute temperature (*Wallace et al., 2002*). Transport of [$^3$H]-glycerol and [$^3$H]-suramin was performed essentially as for [$^3$H]-pentamidine. For [$^3$H]-glycerol (American Radiolabeled Chemicals, 40.0 Ci/mmol), $10^7$ BSF *T. brucei* were incubated with radiolabel at a final concentration of 0.25 μM, for one minute. When the effect of CCP was studied, CCCP was added 3 min prior to the addition of the radiolabel. [$^3$H]-suramin (American Radiolabeled Chemicals, 20.0 Ci/mmol) was also used at 0.25 μM final concentration, using 15 min incubations in the presence and absence of 100 μM unlabelled suramin (used as saturation) control.

## Construction of AQP mutants and transfection

All mutations in the TbAQP2 and TbAQP3 genes were introduced to the relevant backbone WT vector, either pRPa$^{GFP-AQP2}$ or pRPa$^{GFP-AQP3}$ (*Baker et al., 2012*), by site-directed mutagenesis. Use of the pRPa vector for transfection of 2T1-derived *T. brucei* ensures integration in a prepared locus in the ribosomal rRNA spacer region and a high level of stable expression (*Alsford et al., 2005*). For mutations S131P, S263A, I110W, L264R, L258Y, I190T and W192G in AQP2, and W102I, R256L and Y250L in AQP3 mutations were inserted using the QuikChange II kit (Agilent, Santa Clara, CA, USA), following the manufacturer's instructions. For mutations L84W, L118W, L218W, L84M, L118M and L218M were introduced using the Q5 Site-Directed Mutagenesis Kit (E0554S), (New England BioLabs) according to manufacturer's instructions.

The following primer pairs (itemised in *Supplementary file 4*) were used to insert the named TbAQP2 mutations: for S131P, primers HDK1062 and HDK1063; in combination with mutation S263A, using primers HDK1064 and HDK1065 to produce plasmid pHDK166. For I110W, primers HDK607 and HDK608, to produce pHDK84; for L264R, primers HDK609 and HDK610 to produce pHDK167; the combination I110W/L264R was produced using primers HDK609 and HDK610 on plasmid pHDK84 to give plasmid pHDK78; for L258Y, primers HDK1109 and HDK1110 to produce pHDK168; for I190T, primers HDK1056 and HDK1057 to produce pHDK163; for W192G, primers HDK1058 and HDK1059 to produce pHDK164; the combination I190T/W192G was produced using primers HDK1060 and HDK1061 on plasmid pHDK163 to give plasmid pHDK165; for L84W, primers HDK1276 and HDK1277, producing pHDK210; for L118W, primers HDK1274 and HDK1275, producing pHDK208; for L218W, primers HDK1272 and HDK1273, producing pHDK209; for the combination L84W/L118W, primers HDK1276 and HDK1277 on template pHDK208, producing pHDK227; for L84M, primers HDK1364 and HDK1367, producing pHDK234; for L118M, primers HDK1365 and HDK1367, producing pHDK235; and for L218M, primers HDK1366 and HDK1367, producing pHDK236. To insert the named mutations into TbAQP3, the following primers were used (*Supplementary file 5*): for W102I, primers HDK511 and HDK512, in combination with mutation R256L, with primers HDK513 and HDK514, to produce plasmid pHDK71; and to add mutation Y250L to this combination, primers HDK795 and HDK796, to produce pHDK121. All plasmids were checked by Sanger Sequencing (Source BioScience, Nottingham, UK) for the presence of the correct mutation(s) and the cassette for integration digested out with *Asc*I (NEB, Hitchin, UK) prior to transfection.

For transfection, 10 µg of digested plasmid and $1–2 \times 10^7$ parasites of the desired cell line (either *aqp2/aqp3* null or *aqp1/aqp2/aqp3* null) were resuspended in transfection buffer and transfected using an Amaxa Nucleofector, with program X-001. After a recovery period (8–16 hr) in HMI-11 at 37°C and 5% $CO_2$, the parasites were cloned out by limiting dilution with the selection antibiotic (2.5 µg/ml hygromycin). In all cases the presence of the construct and its correct integration into the designed rRNA locus was verified by three PCR reactions, one using primers for the amplification of the full-length aquaporin (primers HDK529 and HDK209). The second PCR was performed to amplify the gene with surrounding parts of the expression cassette using (primers HDK1011 and HDK430). The third PCR was to assess whether the expression cassette had linearized and integrated into the *T. brucei* genome using (primers HDK991 and HDK713).

## Drug sensitivity assays

Drug sensitivity assays for *T. b. brucei* bloodstream forms used the cell viability dye resazurin (Sigma) and were performed exactly as described (*Wallace et al., 2002*; *Bridges et al., 2007*) in 96-well plates with doubling dilutions of test compound, starting at 100 µM, over 2 rows of the plate (23 dilutions plus no-drug control). Incubation time with test compound was 48 hr (37°C/5% $CO_2$), followed by an additional 24 hr in the presence of the dye.

## Protein turnover and western blotting

*T. brucei* 2T1 containing $^{3\times HA}$AQP2 were incubated with 1 µg/ml tetracycline for 24 hr at 37°C/5% $CO_2$ to induce expression. Tetracycline was then washed away by four consecutive washes with fresh supplemented HMI-9, and cells were then incubated with 25 nM of pentamidine ($5 \times EC_{50}$) for 1 hr at 37°C/5% $CO_2$ prior to harvest cells for pulse chase experiments. To determine protein half-life, translation was blocked by addition of cycloheximide (100 µg/ml) and cells were harvested at various times by centrifugation ($800 \times g$ for 10 min at 4°C). Cells were washed with ice-cold PBS, then resuspended in $1 \times$ SDS sample buffer (Thermo) and incubated at 70°C for 10 min. Proteins were subjected to electrophoresis using 4–12% precast acrylamide Bis-Tris gels (Thermo) and transferred to polyvinylidene difluoride (PVDF; Sigma-Aldrich) membranes with a iBlot2 system (Thermo) at 23 V for 6 min, exactly as described (*Quintana et al., 2020*), blocking non-specific binding with 5% (w/v) bovine serum albumin (BSA; Sigma) in Tris-buffered saline (pH 7.4) with 0.2% (v/v) Tween-20 (TBST) and using rat anti-HA IgG$_1$ (Sigma) or anti-mouse β-tubulin (clone KMX-1; Millipore) at 1:5000 or 1:10,000 dilution in TBST, respectively. Membranes were washed five times with TBST and then incubated in TBST/1% BSA with the appropriate horseradish peroxidase (HRP)-coupled secondary antibody (Sigma), at 1: 10,000. Bands were visualised using the ECL Prime Western Blotting System

(Sigma) and GE healthcare Amersham Hyperfilm ECL (GE). Densitometry quantification was conducted using ImageJ software (NIH).

## Molecular dynamics

Molecular dynamics simulations were performed using the GROMACS software package, version 5.1.1 (*Abraham et al., 2015*). We used the coordinates from the homology model of TbAQP2 published in Figure 2A in *Munday et al., 2015a*, which was inserted into POPC/POPE (4:1) membranes, approximately reflecting the membrane composition of *T. b. brucei* (*Smith and Bütikofer, 2010*). The membrane models were constructed using the CHARMM-GUI webserver (*Jo et al., 2008*). Subsequently, extended stability tests of the modelled structure and the bound pentamidine were carried out using unbiased simulations of 100 ns length. The root-mean-square deviation (RMSD) of the protein remained relatively low with a backbone RMSD converging to ~3 Å after 100 ns simulated time (*Figure 8—figure supplement 1*) bound to the binding site defined previously using molecular docking (*Figure 8A*; *Munday et al., 2015a*). For these and all following simulations, we used the CHARMM36 force field (*Klauda et al., 2010*) pentamidine was parameterised using the CHARMM generalized force field approach (CHGenFF [*Vanommeslaeghe et al., 2010*]). All simulations employed a time step of 2 fs under an NPT ensemble at p=1 bar and T = 310 K. To obtain non-equilibrium work values for removing pentamidine from the internal AQP2 binding site, we then conducted steered MD simulations with a probe speed of 0.005 nm/ns and a harmonic force constant of 300 kJ/mol nm$^2$, pulling pentamidine in both directions along the pore axis. The free energy profile of pentamidine binding to the AQP2 pore was reconstructed by using the *Jarzynski, 1997* equality.

## Statistical analysis

All transport experiments were performed in triplicate and all values such as rate of uptake, percent inhibition, $K_i$, $K_m$, $V_{max}$ etc were performed at least three times completely independently. For drug sensitivity tests, all $EC_{50}$ values were based on serial dilutions over two rows of a 96-well plate (23 doubling dilutions plus no-drug control), which were obtained independently at least three times. $EC_{50}$ and $IC_{50}$ values were determined by non-linear regression using the equation for a sigmoid curve with variable slope and are presented as average ± SEM. Statistical significance between any two data points was determined using Student's t-test (unpaired, two-tailed).

## Acknowledgements

This work was supported by the UK Medical Research Council (MRC) [grant G0701258 to HPdK] and by the US National Institutes of Health (NIH) [Grant No. GM111749 to DWB]. DC was supported by an MRC iCASE award [MR/R015791/1]. UZ acknowledges funding from the Scottish Universities Physics Alliance. AHA is funded through a PhD studentship from Albaha University, Saudi Arabia. GDC was funded by a PhD Studentship from Science Without Borders [206385/2014–5, CNPq, Brazil]. TS was funded via a Doctoral Training Programme of the MRC and the Cambridge Trust and SW was funded by a Sir Henry Dale fellowship of the Wellcome Trust and Royal Society. The authors thank Dr Tansy Hammarton for the use of the CRK2 RNAi cell line and Prof David Horn for the use of the aqp1-3 null cell line. This work was supported by a grant from the Wellcome Trust (204697/Z/16/Z to MCF). The authors are grateful to Professor George Diallinas, University of Athens, Greece, for his exceptionally insightful reviewer comments and have adopted several of his arguments in revision.

## Additional information

### Competing interests

Fredrik Svensson: Was an employee of IOTA Pharmaceuticals Ltd at the time. The other authors declare that no competing interests exist.

## Funding

| Funder | Grant reference number | Author |
|---|---|---|
| Medical Research Council | G0701258 | Harry P De Koning |
| National Institutes of Health | GM111749 | David W Boykin |
| Medical Research Council | MR/R015791/1 | Harry P De Koning |
| Scottish Universities Physics Alliance | | Ulrich Zachariae |
| Albaha University | | Ali Alghamdi |
| Science Without Borders, Brazil | 206385/2014-5 | Gustavo Daniel Campagnaro |
| Wellcome | 204697/Z/16/Z | Mark Field |
| Medical Research Council | | Teresa Sprenger |
| Cambridge trust | | Teresa Sprenger |
| Wellcome Trust | | Simone Weyand |
| Royal Society | | Simone Weyand |

The funders had no role in study design, data collection and interpretation, or the decision to submit the work for publication.

## Author contributions

Ali H Alghamdi, Gustavo Daniel Campagnaro, Dominik Gurvic, Chinyere E Okpara, Arvind Kumar, Juan Quintana, Maria Esther Martin Abril, Patrik Milić, Laura Watson, Daniel Paape, Anna Dimitriou, Joanna Wielinska, Graeme Smart, Laura F Anderson, Christopher M Woodley, Siu Pui Ying Kelly, Hasan MS Ibrahim, Mohammed I Al-Salabi, Anthonius A Eze, Teresa Sprenger, Ibrahim A Teka, Simon Gudin, Investigation; Jane C Munday, Fredrik Svensson, Investigation, Methodology; Luca Settimo, Validation, Investigation, Methodology; Fabian Hulpia, Formal analysis, Methodology, Writing - original draft; Simone Weyand, Supervision; Mark Field, Conceptualization, Supervision, Funding acquisition, Methodology; Christophe Dardonville, Resources; Richard R Tidwell, Paul O'Neill, David W Boykin, Resources, Supervision; Mark Carrington, Conceptualization, Investigation, Methodology; Ulrich Zachariae, Formal analysis, Supervision, Methodology; Harry P De Koning, Conceptualization, Data curation, Formal analysis, Supervision, Funding acquisition, Methodology, Writing - original draft, Writing - review and editing

## Author ORCIDs

Gustavo Daniel Campagnaro (iD) http://orcid.org/0000-0001-6542-0485
Fabian Hulpia (iD) http://orcid.org/0000-0002-7470-3484
Anthonius A Eze (iD) http://orcid.org/0000-0002-4821-1689
Mark Carrington (iD) https://orcid.org/0000-0002-6435-7266
Harry P De Koning (iD) https://orcid.org/0000-0002-9963-1827

## Decision letter and Author response

Decision letter https://doi.org/10.7554/eLife.56416.sa1
Author response https://doi.org/10.7554/eLife.56416.sa2

# Additional files

## Supplementary files

- Supplementary file 1. Table of all HAPT/TbAQP2 inhibitors used in this study.

- Supplementary file 2. Table with all sequences of TbAQP2 and the generated mutants thereof. Mutated nucleotides and codons are indicated.

- Supplementary file 3. Synthesis of all new inhibitors used in this study.

- Supplementary file 4. List of primers used in the construction of mutations in TbAQP2.

- Supplementary file 5. Primers used for mutations in TbAQP3.
- Transparent reporting form

## Data availability

All data generated or analysed during this study are included in the manuscript and supporting files. Source data files have been provided that cover all figures and give raw data, averages, statistics etc.

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
