## [Decision Letter]

Thank you for submitting your article "Positively selected modifications in the pore of TbAQP2 allow pentamidine to enter Trypanosoma brucei" for consideration by *eLife*. Your article has been reviewed by three peer reviewers, one of whom is a member of our Board of Reviewing Editors, and the evaluation has been overseen by Dominique Soldati-Favre as the Senior Editor. The following individuals involved in review of your submission have agreed to reveal their identity: George Diallinas (Reviewer #2); Eric Beitz (Reviewer #3).

The reviewers have discussed the reviews with one another and the Reviewing Editor has drafted this decision to help you prepare a revised submission.

Summary:

Mutations of Trypanosoma brucei aquaglyceroporin 2 (AQP2) are known to result in resistance to some anti-trpyanosomal drugs. Although it was originally assumed that AQP2 is a transporter for the compounds, it was recently suggested that instead, AQP2 acts as a receptor and the drugs are subsequently taken up by endocytosis. In this paper, the authors provide various lines of evidence that support the original, transporter, hypothesis.

Essential revisions:

The major revisions are to tone down the claims, since definitive proof is not provided. Both reviewers say this; you need to say it too and remove the emphatic adjectives (see reviewer 3). Since both reviewers are aquaglyyceroporin experts and the reviews contain quite a lot of insights they are included in full.

Reviewer #2:

Summary:

The present manuscript attempts to reject the 'receptor endocytosis' mechanism through the following logic. First, they perform a relative phylogenetic analysis of AQPs that shows that TbAQ2 orthologues have evolved from more canonical AQPs in a way that they possess non-canonical selectivity elements. They propose that these particular selectivity elements of TbAQ2 enlarge the pore size and allow access of pentamidine and other larger cations into the substrate trajectory. Given that the TbAQ3 paralogue, unlike TbAQ2, is not involved in pentamidine uptake, they then try to genetically interconvert the selectivity filters of TbAQ2 and TbAQ3, and test whether this is sufficient to confer gain or loss of sensitivity/resistance to pentamidine.

Their evidence supports that the exchanged selectivity elements does play a role in pentamidine uptake and sensitivity. These results are not discussed further in respect to some specificity changes observed in respect to different drugs (see my specific comment later).

Subsequently, they employ a rational mutational analysis of amino acids modelled to potentially bind pentamidine (and melarsoprol) or act as putative molecular barriers at the cytoplasmic end of the pore (i.e. inner filter or gate). They show that mutations in the putative pentamidine binding site dramatically reduce pentamidine transport and some of them also the sensitivity to the drug, and mutations introducing bulkier than the natural residues at the inner exit of pore also show much reduced rates of pentamidine transport and relative reduced pentamidine sensitivity. Noticeably, mutations reducing pentamidine transport had a minor effect on glycerol efflux, which seems to be the physiological function of TbAQ2. At this point they describe a number of subsequent experiments which, they claim, provide evidence against the involvement of endocytosis in pentamidine uptake (through the use of a strain partially blocked in endocytosis in blood stream forms) and supporting the need for a proton motive force to drive TbAQP2-mediated pentamidine uptake.

What follows is an in silico approach for 'docking' pentamidine in its proposed binding site in the modelled structure of TbAQ2, followed by Molecular Dynamics supporting that pentamidine can exit the channel in either direction. Finally, the authors perform an extensive SAR analysis of the pentamidine-AQP2 interactions, using a plethora of designed and chemically made pentamidine analogues or similar drugs, which shows that both amidine groups and one or both ether oxygens in pentamidine are involved in interactions with AQP2. This analysis, they claim, unambiguously demonstrates pentamidine binding occurs in an elongated orientation and is in complete agreement with the modelling and molecular dynamics, and the mutational analysis presented before.

Essential revisions:

- I will start form the end, saying that the basic conclusion of the authors (TbAQ2 is a specific aquaporin transporting pentamidine), is well justified from their phylogenetic and mutational analyses, modelling, MDs and SAR, as well as, the fact the TbAQ2 is a channel. I am quite convinced that pentamidine uptake is via bona fidae channel activity and not by endocytosis.

- On the other hand, their experiments aiming to show that endocytosis is not involved in pentamidine uptake (part 2) is very weak and not convincing. First of all, the strain used is only moderately blocked in endocytosis and the results they get (Figure 4) concerning the uptake of pentamidine versus suramin are statistically unconvincing.

- Most importantly, drawing conclusions on specific endocytosis of TbAQ2 based on the rate of pentamidine versus suramin uptake is conceptually false. Endocytosis is a cargo-dependent process, a notion that seems to escapes to many, so that different protein cargoes use different mechanism of endocytosis (e.g. clathrin-dependent or clathrin-independent, AP2-dependent or AP2-independent, polarized or non-polarized, cargo conformation-depended or not, etc.).

- If the authors wished to formally dismiss the idea or receptor-endocytosis they should have directly accessed microscopically and by westerns, using GFP-tagging, the internalization/turnover of TbAQP2 in the presence or absence of pentamidine and other drugs or analogues.

- Substrate- or transport activity-dependent endocytosis of transporter is a well-known phenomenon in fungi and seemingly also in plants and mammals (Gournas et al., 2010; Adv Exp Med Biol, 2016; Gormal et al., 2018). Multiple evidence suggests that upon binding of their substrates or ligands transporters undergo conformational changes related to the alternation from an outward to an inward conformation, but also related to the opening and closing of outer or inner gates. In some cases, the entire transport cycle is needed to be complete to elicit endocytosis, but in other cases an intermediate conformation (e.g. inward-facing, substrate occluded conformation) is sufficient to promote endocytosis (Gournas et al., 2010; Ghaddar et al., 2014; Gournas et al., 2017). The generally accepted idea is that a specific substrate/ligand-induced transient conformation make the transporter accessible to ubiquitination and thus to endocytosis. What is important to realize is that it is not possible to predict a priori whether substrate binding is sufficient to trigger endocytosis, or whether a full transport cycle in needed, or even cases where substrates do not cause endocytosis. Each transporter might behave differently in respect to endocytosis by substrates or ligands, particularly under distinct physiological, developmental or stress conditions, and endocytosis of a specific transporter should not be related with endocytosis of other cargoes or general endocytic activity, without targeted experiments.

- If the authors have performed the relative endocytosis studies on TbAQ2 they would have known whether pentamidine or other drugs elicit endocytosis. In case pentamidine elicits endocytosis this can still be unrelated to the mechanism of pentamidine uptake, as several transporters are endocytosed by their substrates. If a TbAQ2 endocytosis serves accumulation of pentamidine this would, in principle, be a fast process. In contrast, in cases endocytosis is the result a feedback mechanism for controlling excess substrate accumulation, this usually is slower becoming visible after a significant time period of transport activity. Moreover, in case pentamidine does not elicit internalization of TbAQ2, then they could easily dismiss endocytosis as the mechanism of uptake. I also suggest to the authors that if they wish to do these experiments to use the procyclic forms of T. brucei, if possible, where TbAQ2 is localized all along the PM and it will thus be easier to detect and quantify endocytosis microscopically and relate it to transport activity measurements., In any case, I propose to remove (or transfer to supplementary material) their experiments describe in part 2. In fact, the authors give convincing explanations in their discussion why it is difficult to reconcile the literature on pentamidine transport/resistance with uptake via endocytosis (Discussion section).

- I also propose to the authors to remove the experiments on the role of the proton motive force in AQP2-mediated pentamidine uptake, as CCCP has both pleiotropic and specific effects on TbAQ2, difficult to put under a rigorous conclusion

- An information I could not find in the present manuscript or in Song et al., (2016) is whether pentamidine can inhibit competitively (or non-competitively) TbAQ2-mediated H3-glycerol uptake/efflux. I think this is a very critical experiment, which together with a direct assessment of endocytosis of TbAQ2 and TbAQ3 (as a control) by pentamidine, will be critical in formally showing that TbAQ2 functions as a channel rather that a receptor for the uptake of pentamidine.

In order to be able to have an opinion on the alternative hypothesis of receptor endocytosis-mediated pentamidine uptake, I also read the Song et al., (2016) article. The evidence that TbAQP2 does not transport pentamidine when expressed in yeast is indirect through growth tests and is not convincing (they have not tested directly uptake of radiolabeled pentamidine). In addition, there are no data showing that TbAQ2 is properly sorted to the PM of yeast (the western shown lacks controls). This would need functional tagging of TbAQ2 with GFP. Moreover, expression in yeast might not reconstitute full activity and physiological kinetics. The aforementioned problems are very well known bottlenecks in efforts to express transporters in heterologous systems. Furthermore, a main criterion used by Song et al. is that "if a solute is excluded (e.g. formamidine) from passing, then a much larger (pentamidine) would be excluded as well". This is surely a misconception, well known to people working on the molecular basis of transporter specificity. Finally, Song et al., argues that since in BSF TbAQ2 localizes in the trypanosome flagellar pocket, which is a site of very efficient exocytosis and endocytosis, then this is compatible with the idea of endocytosis-mediated pentamidine uptake. As discussed before general endocytosis cannot be related with endocytosis if a specific transporter without relative experimental evidence. Overall, the idea of Song et al., that pentamidine-induced endocytosis of TbAQP2 acts as vehicle for uptake seems to me purely speculative and rather unlikely.

The phylogenetic and mutational analyses, modelling, MDs and SAR approaches support the most logical scenario, that pentamidine is transported via a channel and not via receptor endocytosis. AQP2 is a channel and I find no rigorous evidence by Song et al., (2016) that it can function as receptor, nor that it is endocytosed in the presence of pentamidine. Furthermore, the criterion of size in respect to specificity in transport via a channel or transporter does not hold true. Specificity cannot be predicted a priori as it depends not only on elements of substrate binding site, but also on major conformational changes and the dynamics of gating, all triggered by substrate binding and release. Several examples of 'surprising' specificities exist in the literature. For example, specific bacterial, fungal or plant NAT transporters can translocate with very high affinity xanthine and uric acid, but not hypoxanthine, which is a very similar solute of smaller in size (Gournas et al., 2008). Also, a fungal NAT transporter specific for xanthine/uric acid can transport with extremely high-affinity the drug allopurinol (a hypoxanthine analogue), but surprisingly, transport of allopurinol does not follow Michelis--Menten kinetics (Diallinas and Biochimie, 2013). Finally, NCS1 or APC transporters with practically identical substrate binding site residues transport distinct substrates via the involvement of distinct selective outer an inner gates (Krypotou et al., 2015 Sioupouli et al., 2017; Gournas, 2016).

The authors also give convincing explanations why is difficult to reconcile the literature on pentamidine transport/resistance with uptake via endocytosis in the Discussion section.

- However, if they wanted to formally reject the scenario of endocytosis they should have directly accessed microscopically and by westerns, using GFP-tagging, the internalization/turnover of AQP2 in tricyclics, in the presence or absence of pentamidine and other drugs or analogues, where the channel is evenly localized in the PM and is more active.

The present work can be published in *eLife* after a major revision addressing some of the above points.

Reviewer #3:

The presented multi-author study evaluates experimental data and arguments as a basis to decide whether uptake of the anti-trypanosomal agent pentamidine occurs by passing the channel of an aquaporin isoform, TbAQP2, or by endocytosis of a TbAQP2-pentamidine complex. To this end, the authors generated a number of point mutations of TbAQP2 and of a second isoform, TbAQP3; assayed the effect of ionophores; partially blocked endocytosis by a knock-down approach; simulated docking and molecular dynamics; and synthesized numerous pentamidine-related compounds for structure-activity relationships. Quantitative data derive from radiolabeled pentamidine, effective concentrations (EC50) on parasite growth, and binding competition experiments (K_i), or were computed from the simulations. A tremendous amount of work went into the study.

- The major hurdle to ultimately answer the question seems to be the extremely low rate at which pentamidine is transported into the cells. The authors provide a number in Figure 4B, i.e. about 6 fmols per 10^7 cells per second, which translates into about 23 molecules per cell per second.

- Later, in the Discussion section, the authors state a number from another reference of 9.5 x 10^5 per cell per hour, which is about one order of magnitude higher (about 260 molecules per cell per second). If one guesses the number of aquaporin proteins in a cell to be 1000, which is probably too low as erythrocytes were estimated to contain 200,000 copies, this would mean that the uptake of one pentamidine molecule takes about 50 seconds (or even longer if more than 1000 aquaporins are present).

- There are several references in the manuscript to a study with yeast-expressed TbAQP2 showing inhibition of glycerol permeability by pentamidine (Song et al.). This setup would not pick up such slow transport rates, and, consequently, pentamidine would act as an inhibitor. This would also be true for other cations tested in the yeast. Water permeability of aquaporins is in the 1-nanosecond range and glycerol is slower by two orders of magnitude; hence, there is still a gap of 8 orders of magnitude compared to pentamidine uptake rates into trypanosomes. Another consequence of the long residence time of pentamidine at the aquaporin is that both processes, i.e. membrane-potential driven leakage of pentamidine through the aquaporin and internalisation of the complex, may jointly contribute to uptake (taking into account the numbers obtained for suramin endocytosis in Figure 4C).

1) Results section: in several instances, small differences in the relative uptake numbers are observed for the various TbAQP2 mutants, but potential differences in expression levels or available copies at the plasma membrane are not considered. Measured effects are not necessarily directly connected to the mutation.

2) Subsection “1.4 Mutations of amino acids modelled to potentially bind pentamidine or melarsoprol dramatically reduce pentamidine transport”, and Figure 2A: Presence and absence of significance in the assays using TbAQP2 I190T and TbAQP2 I190T/W192G do not derive from differences in their activity (equal numbers) but from the background level that varies between 0.4% and 1%. This variation lets one wonder if the indicated significance in Figure 1G is true (background of 0.5%; at 1% the stated difference would not be significant).

- Sounds like there is an issue here. Maybe modify conclusion.

3) Subsection “5.5. Non-diamidine trypanocides”, and Figure 9—figure supplement 1: Calculating a resistance factor appears reasonable to estimate the proportion of inhibitor in the cytosol. However, it is questionable whether the comparison of resistance factors is meaningful if the EC50 values differ largely between different compounds. This alone may well explain the lack of correlation.

Therefore, please add the absolute values EC50_TbAQP2/3null and EC50_TbAQP2wt to the table in Figure 9—figure supplement 1. The data do not seem to allow one to draw conclusions on the transport-vs-endocytosis question, see e.g. compounds R14 and R52. R14 exhibits almost equal affinity to TbAQP2, binding of R52 is 200-fold weaker, yet R52 similarly depends on the presence of TbAQP2 as pentamidine, whereas R14 is equally effective with or without TbAQP2. This does neither support transport nor endocytosis.

4) General: the authors quite often use strong wording ("unequivocally", "greatly" etc.) to make their case. This should be toned down a bit as all presented evidence on pentamidine transport is indirect. This is not a shortcoming of the study but inherent to the problem. It would be desirable to have direct, unequivocal transport assays for such very slow events in the future.

---

## [Author Response]

Reviewer #2:Essential revisions:- I will start form the end, saying that the basic conclusion of the authors (TbAQ2 is a specific aquaporin transporting pentamidine), is well justified from their phylogenetic and mutational analyses, modelling, MDs and SAR, as well as, the fact the TbAQ2 is a channel. I am quite convinced that pentamidine uptake is via bona fidae channel activity and not by endocytosis.- On the other hand, their experiments aiming to show that endocytosis is not involved in pentamidine uptake (part 2) is very weak and not convincing. First of all, the strain used is only moderately blocked in endocytosis and the results they get (Figure 4) concerning the uptake of pentamidine versus suramin are statistically unconvincing.

We certainly acknowledge that this experiment is no definitive proof and is not as statistically strong as we would have liked. The problem is that a more complete disabling/inhibition of endocytosis in Trypanosoma brucei bloodstream forms is very quickly lethal. The rate of exocytosis remains very high, leading to a rapid enlargement of the flagellar pocket (FP), which is the only area of the cell where endocytosis and exocytosis takes place and can take place. This in turn leads to gross distortions of the cell shape, and many cellular pathologies, which would invalidate any conclusions drawn from the experiment. We thus had no choice but to go for as early a time point after RNAi induction as allows any observation of diminished endocytosis. This is inherent to the organism being studied. Regardless, we believe the experiment is valid as far as it goes: an indication that endocytosis from the FP is inhibited (^3^H-suramin control, *P*= 0.019) but pentamidine uptake is not. The level of reduction for suramin uptake is consistent with the published effects of CRK12 RNAi at 12 h in the reference given (Monnerat et al., 2013), and the reason the 12 h point was chosen. Still, we have moved the figure to the Supplemental data and revised any statements in the Discussion section that give too much weight to the statement.

- Most importantly, drawing conclusions on specific endocytosis of TbAQ2 based on the rate of pentamidine versus suramin uptake is conceptually false. Endocytosis is a cargo-dependent process, a notion that seems to escapes to many, so that different protein cargoes use different mechanism of endocytosis (e.g. clathrin-dependent or clathrin-independent, AP2-dependent or AP2-independent, polarized or non-polarized, cargo conformation-depended or not, etc.).

There is quite solid evidence that there is no AP-2 dependent endocytosis in trypanosomes, and that endocytosis is clathrin-dependent (Allen et al., 2003; Morgan et al., 2002, 2003). Profs. Mark Field and Mark Carrington are the leading experts on this and co-authors of the current submission. While we agree with the reviewer that we cannot formally rule out that the receptor for suramin (ISG75) and TbAQP2 are endocytosed via a similar mechanism, there is nothing in the extensive literature on this subject that suggests multiple, independent-acting endocytotic mechanisms being employed by *T. brucei* bloodstream forms. The suramin receptor ISG75 and TbAQP2 are both ubiquitinated and directed to the lysosomes (Zoltner et al., 2015; Quintana et al., 2020).

- Substrate- or transport activity-dependent endocytosis of transporter is a well-known phenomenon in fungi and seemingly also in plants and mammals (Gournas et al., 2010; Adv Exp Med Biol, 2016; Gormal et al., 2018). […] Each transporter might behave differently in respect to endocytosis by substrates or ligands, particularly under distinct physiological, developmental or stress conditions, and endocytosis of a specific transporter should be related with endocytosis of other cargoes or general endocytic activity, without targeted experiments.- If the authors have performed the relative endocytosis studies on TbAQ2 they would have known whether pentamidine or other drugs elicit endocytosis. […] In fact, the authors give convincing explanations in their discussion why it is difficult to reconcile the literature on pentamidine transport/resistance with uptake via endocytosis (Discussion section).

We genuinely thank the reviewer for his expert comments and insights. One reason we did not include direct endocytosis of aquaporin experiments in our manuscript was that this was being pursued in parallel, in collaboration, by Prof. Mark Field and his team, and constitutes a separate and challenging project, now accepted for publication (Quintana et al., 2020). Prof. Field is the acknowledged expert on trafficking of membrane proteins in trypanosomes. There are two reasons the suggestion of the reviewer to use procyclic cells for such experiments was not done. First, the relevant system to look at drug resistance mechanisms is the clinical form of *T. brucei*, the bloodstream form. More important from a cell biology point of view, is that, while the TbAQP2 protein is indeed spread out over the plasma membrane of procyclic *T. brucei*, it can only be internalised from the flagellar pocket, as stated in the Discussion section: “Moreover, in procyclic cells TbAQP2 is spread out over the cell surface (Baker et al., 2012) but endocytosis happens exclusively in the flagellar pocket (Field and Carrington, 2009) (which is 3-fold smaller in procyclic than in bloodstream forms (Demmel et al., 2014)), as the pellicular microtubule networks below the plasma membrane prevent endocytosis (Zoltner et al., 2016).” This makes it actually a less effective form to study TbAQP2 endocytosis, as only a small proportion of the protein is available for endocytosis. But we make the point that, despite most of procyclic TbAQP2 being unavailable for endocytosis, the rate of HAPT1/AQP2-mediated ^3^H-pentamidine uptake is ~10-fold higher in the procyclic form: “..the rate of endocytosis in bloodstream trypanosomes is much higher than in the procyclic lifecycle forms (Langreth and Balber, 1975; Zoltner et al., 2016), yet the rate of HAPT-mediated [^3^H]-pentamidine uptake in procyclics is ~10-fold higher than in bloodstream forms (De Koning, 2001a; Teka et al., 2011), despite the level of TbAQP2 expression being similar in both cases (Siegel et al., 2010; Jensen et al., 2014).”

Importantly, the paper on TbAQP2 endocytosis from the Mark Field laboratory has now been accepted for publication (Quintana et al., 2020). This paper shows that TbAQP2, like ISG75 (the suramin receptor), is ubiquitylated and that this is essential for its stability and for internalisation and trafficking – making it even more likely that suramin and pentamidine are endocytosed using similar mechanisms. The paper also showed ISG75 and TbAQP2 co-localisation by immunocytochemistry. Another noteworthy discovery reported in that paper is that TbAQP2 exists in the trypanosome as a tetramer-of-tetramers, making it highly unlikely that the AQP2 protein is able to undergo substantial conformational changes – nor are these known from aquaporins in general. Finally, the paper shows that TbAQP2 has a half-life time >4 hours, which seems completely incompatible with receptor-mediated endocytosis of pentamidine when the uptake must be measured in a scale of seconds (as opposed to [^3^H]-suramin, 15 minutes to get a reliable signal).

The reviewer asks us to include an assessment of the effect of pentamidine on the rate of turnover of TbAQP2. With the Quintana et al., paper now in press and available online we can include this experiment, which had already been performed, as new Figure 4. The figure shows the rate of turnover of AQP2, as quantified by Western blot after cycloheximide pre-treatment. The original blots are included in the supplemental materials. At the concentration of pentamidine used, close to the K_m_ and 5× the EC_50_ value, the drug had no effect at all on the turnover rate of TbAQP2. This information was not included in the Quintana et al., manuscript, and because of this contribution Juan Quintana and Mark Field have been added to the author list.

In conclusion, we believe that, in the revised version, there is very substantial though arguably not conclusive proof arguing against a link between TbAQP2 endocytosis and pentamidine uptake. We have, as requested, qualified the conclusions to reflect that these various observations individually should not be considered as definitive proof. However, we do believe that the totality of evidence and the number of unrelated, independent approaches together, do make a very compelling argument that pentamidine is taken up through the pore rather than via endocytosis.

- I also propose to the authors to remove the experiments on the role of the proton motive force in AQP2-mediated pentamidine uptake, as CCCP has both pleiotropic and specific effects on TbAQ2, difficult to put under a rigorous conclusion

Again, we agree with the reviewer that these results on their own do not constitute complete proof of either model. However, the results with the ionophores nigericin and gramicidin do establish that the plasma membrane potential V_m_ is essential to drive the pentamidine into the cell, in close agreement with the molecular dynamics modelling, our previous results with [^3^H]-pentamidine transport obtained in procyclic *T. brucei* (De Koning, 2001), and the observation that knockdown of plasma membrane proton ATPases leads to reduced sensitivity to cationic pentamidine but not to neutral arsenicals (Alsford et al., 2012; Baker et al., 2013). Even without CCCP there is a clear correlation between [^3^H]-pentamidine uptake rate and the protonmotive force (PMF) (Figure 5C). As to CCCP we realised and acknowledged the pleiotropic effects of direct inhibition of TbAQP2 and we believe that this is important to note, but the correlation with PMF is not dependent on the inclusion of CCCP in the dataset, neither in this paper nor in the previous paper on pentamidine uptake in procyclic cells (De Koning, 2001). Given the importance of the ionophore results for the molecular dynamics, and for the interpretation of the published observations regarding the HA1-3 proton ATPases, we believe the experiments with ionophores are important for the details (“loose ends”, if you will) of pentamidine permeation into trypanosomes, rather than for the principal evaluation of permeation model, and we would prefer to keep them in the manuscript.

- An information I could not find in the present manuscript or in Song et al., (2016) is whether pentamidine can inhibit competitively (or non-competitively) TbAQ2-mediated H3-glycerol uptake/efflux. I think this is a very critical experiment, which together with a direct assessment of endocytosis of TbAQ2 and TbAQ3 (as a control) by pentamidine, will be critical in formally showing that TbAQ2 functions as a channel rather that a receptor for the uptake of pentamidine.

The information on pentamidine inhibition of [^3^H]-glycerol uptake is in fact included in the manuscript. Figure 5 frames D and E compare side-by-side the effects of pentamidine (and CCCP) on the uptake of [^3^H]-glycerol and [^3^H]-pentamidine, showing, among other things, that pentamidine does inhibit glycerol flux though TbAQP2, with an EC_50_ value (27.5 nM, average of 2 exp.; the CCCP EC_50_ was n=3 but only two pentamidine EC_50_ values were obtained and as such the value was not included in the manuscript) that is similar to that of its inhibition of [^3^H]-pentamidine uptake. This had not been explicitly drawn attention to in the text, hence it was overlooked; this has now been remedied. But I am not sure how the observation distinguishes between the channel and endocytosis models, which is why we did not use it in the Discussion section of the model. Incidentally, up to 1 mM glycerol does not inhibit [^3^H]-pentamidine uptake, which we attribute to the vast difference in binding affinity and the fleeting association of glycerol with the channel-lining amino acids.

In order to be able to have an opinion on the alternative hypothesis of receptor endocytosis-mediated pentamidine uptake, I also read the Song et al., (2016) article. […] Overall, the idea of Song et al., that pentamidine-induced endocytosis of TbAQP2 acts as vehicle for uptake seems to me purely speculative and rather unlikely.

We thank the reviewer for his comments. We have made similar points in the Discussion section, particularly concerning the notion of size exclusion.

The phylogenetic and mutational analyses, modelling, MDs and SAR approaches support the most logical scenario, that pentamidine is transported via a channel and not via receptor endocytosis. […] Finally, NCS1 or APC transporters with practically identical substrate binding site residues transport distinct substrates via the involvement of distinct selective outer an inner gates (Krypotou et al., 2015 Sioupouli et al., 2017; Gournas, 2016).The authors also give convincing explanations why is difficult to reconcile the literature on pentamidine transport/resistance with uptake via endocytosis in the Discussion section.- However, if they wanted to formally reject the scenario of endocytosis they should have directly accessed microscopically and by westerns, using GFP-tagging, the internalization/turnover of AQP2 in tricyclics, in the presence or absence of pentamidine and other drugs or analogues, where the channel is evenly localized in the PM and is more active.The present work can be published in eLife after a major revision addressing some of the above points.

We thank the reviewer for his considerable input and constructive, fair criticisms. We have certainly taken the revision very seriously and believe we have been able to accommodate the reviewer’s directions.

Reviewer #3:- The major hurdle to ultimately answer the question seems to be the extremely low rate at which pentamidine is transported into the cells. The authors provide a number in Figure 4B, i.e. about 6 fmols per 10^7 cells per second, which translates into about 23 molecules per cell per second.- Later, in the Discussion section, the authors state a number from another reference of 9.5 x 10^5 per cell per hour, which is about one order of magnitude higher (about 260 molecules per cell per second).

The number in Figure 4B was obtained with the CRK12 RNAi line obtained from Dr Tansy Hammarton in our Institute, whereas the published V_max_ value was obtained in 2001 using our own clonal line of Tbb Lister s427. More to the point, the Vmax value is of course the *maximum rate* at saturation concentration of [^3^H]-pentamidine, whereas the rate of uptake in Figure 4B was obtained with a label concentration of just 25 nM, i.e. a concentration below the K_m_, as is good practice. Therefore, it is to be expected that this rate is considerably lower than the V_max_.

If one guesses the number of aquaporin proteins in a cell to be 1000, which is probably too low as erythrocytes were estimated to contain 200,000 copies, this would mean that the uptake of one pentamidine molecule takes about 50 seconds (or even longer if more than 1000 aquaporins are present).

I am not sure that I follow this calculation or can discern how it is relevant. Yes, the rate of uptake can be expressed as 265 molecules per cell per s. Thus, if 1000 AQP units are present at the cell surface at any given time it seems to me that each takes up a pentamidine every 3.8 (i.e, 265/1000) seconds, not 50. All this means is that, the small substrates such as water and glycerol, that use TbAQP2 as a channel, have a much faster permeation rate, which is entirely as expected. The difference in substrate affinity is phenomenal, with the K_m_ of pentamidine as low as ~35 nM, compared to an affinity for glycerol that is no-doubt in the millimolar range, so it is expected that the off-rate for pentamidine is very much slower than for glycerol. And yes, that means that pentamidine does occlude the channel for what is a relatively long period, thereby inhibiting glycerol passage as we show, yet it also does penetrate into the cell, at a sufficient rate to have a pharmacological effect.

For the endocytosis route, of course, the rate of uptake would be even much slower. As stated in the Discussion section, “given a 1:1 stoichiometry for AQP2:pentamidine the endocytosis model would require the internalisation and recycling of as many units of TbAQP2 and this seems unlikely”. If we use the t_1/2_ for TbAQP2 of Quitana et al., (2020) of 4 h, the uptake of 265 molecules of pentamidine per second would require a presence of 3.8×10^6^ copies of TbAQP2 on the cell surface, all in the flagellar pocket of a single cell – an impossibility in my view [9.54×10^5^ molecules per cell per h transported × 4 h]. And again, in procyclic cells the V_max_ is ~10-fold higher despite a much lower endocytosis rate. These further considerations have been added to the Discussion section.

- There are several references in the manuscript to a study with yeast-expressed TbAQP2 showing inhibition of glycerol permeability by pentamidine (Song et al.). This setup would not pick up such slow transport rates, and, consequently, pentamidine would act as an inhibitor. This would also be true for other cations tested in the yeast. Water permeability of aquaporins is in the 1-nanosecond range and glycerol is slower by two orders of magnitude; hence, there is still a gap of 8 orders of magnitude compared to pentamidine uptake rates into trypanosomes. Another consequence of the long residence time of pentamidine at the aquaporin is that both processes, i.e. membrane-potential driven leakage of pentamidine through the aquaporin and internalisation of the complex, may jointly contribute to uptake (taking into account the numbers obtained for suramin endocytosis in Figure 4C).

These issues are partly already addressed in the previous point. Again I am not sure where the reviewer is going with his comment on the yeast-expressed TbAQP2, or how it is relevant to the interpretation of our results presented in the current manuscript, using [3H]-pentamidine as direct measurement, in the original cells. We are happy to concede that the total accumulation of pentamidine is a combination of direct permeation and endocytosis, it was never denied, and has been further added explicitly to the Discussion section. However, we remain convinced, based on the evidence and reasoning presented that by far the larger of these two is permeation through the channel: “Although it is likely that AQP2-bound pentamidine is internalised as part of the natural turnover rate of the protein, this is not likely to contribute very significantly to the overall rate of uptake of this drug.”

1) Results section: in several instances, small differences in the relative uptake numbers are observed for the various TbAQP2 mutants, but potential differences in expression levels or available copies at the plasma membrane are not considered. Measured effects are not necessarily directly connected to the mutation.

This is certainly true, and just measuring expression levels by qRT-PCR, or Northern blot would not necessarily give a quantification of actual TbAQP2, nor establish whether that protein is functionally present in the flagellar pocket. Moreover, The constructs were N-terminally GFP-tagged but Western blots and immunofluorescence microscopy are not quantitative enough to account for small changes in sensitivity and could also not be used as controls; Western blots would also not provide information about the cellular localisation of the protein. Incidentally, Baker et al., (2012) already investigated carefully whether N-terminal tagging of TbAQP2 and TbAQP3 affects their functionality with respect to pentamidine and melarsoprol sensitivity and showed it does not.

Thus, such controls could properly only be functional, which we did by showing functional expression quantified by the EC_50_ of TAO inhibitor SHAM and the uptake rate of [^3^H]-glycerol. While further limitations of that approach can also be identified, this really is the best way this could have been done. Moreover, for most of the mutants the changes in transport were not “small differences”, and gave a very consistent set of results, including the reciprocity of the swap of ar/R amino acids between TbAQP2 and TbAQP3, which lead to a gain of pentamidine transport function for the altered TbAQP3.

2) Subsection “1.4 Mutations of amino acids modelled to potentially bind pentamidine or melarsoprol dramatically reduce pentamidine transport”, and Figure 2A: Presence and absence of significance in the assays using TbAQP2 I190T and TbAQP2 I190T/W192G do not derive from differences in their activity (equal numbers) but from the background level that varies between 0.4% and 1%. This variation lets one wonder if the indicated significance in Figure 1G is true (background of 0.5%; at 1% the stated difference would not be significant).- Sounds like there is an issue here. Maybe modify conclusion.

Yes, the level of pentamidine uptake in the TbAQP2^I190T^ and TbAQP2^I190T/W192G^ mutants is very low, approximately 2.5% of the AQP2-WT control in both cases, as depicted in Figure 2A. It is also agreed on that because of the difference in the aqp2/aqp3 null controls performed in parallel with each experiment, and thus separately for the two mutants, happens to result in the P>0.05 threshold difference from the *null* control being reached for TbAQP2^I190T^ but not for TbAQP2^I190T/W192G^ (P=0.18, n=4, unpaired t-test), all as indicated in the Figure. The issue here, in any case, is not whether pentamidine transport by the mutants was significantly different from the null background, but from the WT AQP2 control, which it was, again as indicated. As to whether the acknowledged inter-assay variation of pentamidine accumulation in the null mutant, when expressed as percent of WT control, between 0.5 and 1.0%, this is precisely why it is essential to perform the same experiment with the *null* control in parallel, with cultures grown in parallel to the same density and using the same solutions, buffers and radiolabel cocktail on the day, every single time; the same goes for the WT control. As such, having done so, the data presented in Figure 1G can be taken as genuine. In three independent experiments, each in triplicate) the [^3^H]-pentamidine transport rate of the triple mutant was approximately double the rate of the null control or single mutant, resulting in P<0.01 in an unpaired test whether percentage of control or pmol/10^7^ cells/s^-1^ was used. It is correct that the greatest care is required when interpreting small differences like this but we believe we have done so to the maximum extent possible.

3) Subsection “5.5. Non-diamidine trypanocides”, and Figure 9—figure supplement 1: Calculating a resistance factor appears reasonable to estimate the proportion of inhibitor in the cytosol. However, it is questionable whether the comparison of resistance factors is meaningful if the EC50 values differ largely between different compounds. This alone may well explain the lack of correlation.Therefore, please add the absolute values EC50_TbAQP2/3null and EC50_TbAQP2wt to the table in Figure 9—figure supplement 1. The data do not seem to allow one to draw conclusions on the transport-vs-endocytosis question, see e.g. compounds R14 and R52. R14 exhibits almost equal affinity to TbAQP2, binding of R52 is 200-fold weaker, yet R52 similarly depends on the presence of TbAQP2 as pentamidine, whereas R14 is equally effective with or without TbAQP2. This does neither support transport nor endocytosis.

The requested data has been added to as an extended Table to Figure 9—figure supplement 1. The reviewer will see that the majority of the compounds displayed sub-μM activity against *T. brucei* WT (2T1 strain) although there were some with lower activity. These were included as the HAPT K_i_ values were available, determined because the structure was of interest for the SAR analysis. However, I am not sure what the reviewer means by “it is questionable whether the comparison of resistance factors is meaningful if the EC_50_ values differ largely between different compounds. This alone may well explain the lack of correlation.” We believe that for a correlation plot such as Figure 9—figure supplement 1 it is in fact necessary to cover a range of values on both axes: the more diverse the dataset is, the better. and there is a substantial range and diversity in both of the parameters plotted against each other. I am genuinely at a loss to answer the query. However, we have already adapted the text around this figure in response to reviewer 2 (see above) and hope that the reviewer will be satisfied with the new phrasing.

4) General: the authors quite often use strong wording ("unequivocally", "greatly" etc.) to make their case. This should be toned down a bit as all presented evidence on pentamidine transport is indirect. This is not a shortcoming of the study but inherent to the problem. It would be desirable to have direct, unequivocal transport assays for such very slow events in the future.

We have done as requested and removed / toned done the adjectives. Nonetheless, we stand by our assertion that, taken all together, we provide strong evidence through many independent methodologies, for the model that pentamidine does pass through the channel of TbAQP2. As to the desirability of measuring ‘such slow events’, we believe we have done exactly that, using custom-made radiolabelled pentamidine of the highest specific activity that was possible. Our method of measuring pentamidine transport has been very accurate and highly reproducible.